# Patient-specific genomics and cross-species functional analysis implicate LRP2 in hypoplastic left heart syndrome

Jeanne L Theis[1†], Georg Vogler[2†], Maria A Missinato[2†], Xing Li[3], Tanja Nielsen[2,4], Xin-Xin I Zeng[2], Almudena Martinez-Fernandez[5], Stanley M Walls[2], Anaïs Kervadec[2], James N Kezos[2], Katja Birker[2], Jared M Evans[3], Megan M O'Byrne[3], Zachary C Fogarty[3], André Terzic[5,6,7,8], Paul Grossfeld[9,10], Karen Ocorr[2], Timothy J Nelson[6,7,8‡*], Timothy M Olson[5,6,8‡*], Alexandre R Colas[2‡], Rolf Bodmer[2‡*]

[1]Cardiovascular Genetics Research Laboratory, Rochester, United States; [2]Development, Aging and Regeneration, Sanford Burnham Prebys Medical Discovery Institute, La Jolla, United States; [3]Division of Biomedical Statistics and Informatics, Mayo Clinic, Rochester, United States; [4]Doctoral Degrees and Habilitations, Department of Biology, Chemistry, and Pharmacy, Freie Universität Berlin, Berlin, Germany; [5]Department of Cardiovascular Medicine, Mayo Clinic, Rochester, United States; [6]Department of Molecular and Pharmacology and Experimental Therapeutics, Mayo Clinic, La Jolla, United States; [7]Center for Regenerative Medicine, Mayo Clinic, Rochester, United States; [8]Division of Pediatric Cardiology, Department of Pediatric and Adolescent Medicine, Mayo Clinic, Rochester, United States; [9]University of California San Diego, Rady's Hospital, San Diego, United States; [10]Division of General Internal Medicine, Mayo Clinic, Rochester, United States

*For correspondence:
nelson.timothy@mayo.edu (TJN);
olson.timothy@mayo.edu (TMO);
rolf@sbpdiscovery.org (RB)

[†]These authors contributed equally to this work
[‡]These authors also contributed equally to this work

Competing interests: The authors declare that no competing interests exist.

**Abstract** Congenital heart diseases (CHDs), including hypoplastic left heart syndrome (HLHS), are genetically complex and poorly understood. Here, a multidisciplinary platform was established to functionally evaluate novel CHD gene candidates, based on whole-genome and iPSC RNA sequencing of a HLHS family-trio. Filtering for rare variants and altered expression in proband iPSCs prioritized 10 candidates. siRNA/RNAi-mediated knockdown in healthy human iPSC-derived cardiomyocytes (hiPSC-CM) and in developing *Drosophila* and zebrafish hearts revealed that LDL receptor-related protein *LRP2* is required for cardiomyocyte proliferation and differentiation. Consistent with hypoplastic heart defects, compared to parents the proband's iPSC-CMs exhibited reduced proliferation. Interestingly, rare, predicted-damaging LRP2 variants were enriched in a HLHS cohort; however, understanding their contribution to HLHS requires further investigation. Collectively, we have established a multi-species high-throughput platform to rapidly evaluate candidate genes and their interactions during heart development, which are crucial first steps toward deciphering oligogenic underpinnings of CHDs, including hypoplastic left hearts.

## Introduction

Hypoplastic left heart syndrome (HLHS) is a congenital heart disease (CHD) characterized by under-development of the left ventricle, mitral and aortic valves, and aortic arch. Variable phenotypic manifestations and familial inheritance patterns, together with the numerous studies linking it to a diverse array of genes, suggest that HLHS is genetically heterogeneous and may have significant

environmental contributors (*Elliott et al., 2003*; *Iascone et al., 2012*; *Theis et al., 2015a*; *Theis et al., 2015b*). In this scenario, synergistic combinations of filtering and validating approaches are necessary to prioritize candidate genes and gene variants that may affect cardiogenic pathways throughout the dynamic process of human heart development.

Although the cellular mechanisms for HLHS remain poorly characterized, a recent study reported generation of the first animal model of HLHS. Based on a digenic mechanism, mice deficient for HDAC-associated protein-encoding *Sap130* and protocadherin-coding *Pcdha9* exhibited left ventricular (LV) hypoplasia that was likely due – at least in part – to defective cardiomyocyte proliferation and differentiation, and increased cell death (*Liu et al., 2017*). Similarly in humans, *Gaber et al., 2013* provide evidence that HLHS-LV samples have more DNA damage and senescence with cell cycle arrest, and fewer cardiac progenitors and myocytes than controls. These observations suggest that impaired cardiomyocyte proliferation could be a mechanism contributing to HLHS pathogenesis, although pathogenic genes controlling this process in humans remain to be identified and validated. Therefore, new synergistic experimental approaches are needed to functionally evaluate gene candidates potentially involved in defective cardiogenesis to serve as a platform for probing the postulated oligogenic basis of CHDs, such as HLHS (*Hinton et al., 2007*; *McBride et al., 2005*).

Over the last decade, induced pluripotent stem cells (iPSCs) have provided a revolutionary experimental tool to reveal aspects of the cellular manifestations associated with disease pathogenesis (*Matsa et al., 2016*; *Mercola et al., 2013*; *Moretti et al., 2013*). Progress in next-generation sequencing technology allows rapid whole-genome DNA and RNA sequencing, thereby providing access to high-resolution and personalized genetic information. However, the interpretation of patient-specific sequence variants is often challenged by uncertainty in establishing a pathogenic link between biologically relevant variant(s) and a complex disease (*Cooper and Shendure, 2011*).

Testing numerous potential human disease-susceptibility genes in a mammalian in vivo model has been challenging because of high costs and low throughput. *Drosophila* with its genetic tools has emerged as the low cost, high-throughput model of choice for human candidate disease gene testing, including neurological and cardiac diseases (*Fink et al., 2009*; *Ocorr et al., 2014*; *Şentürk and Bellen, 2018*; *Schroeder et al., 2019*; *Vissers et al., 2020*; *van der Harst et al., 2016*). *Drosophila* has been established as an efficient model system to identify key genes and mechanisms critical for heart development and function that served as prototypes for vertebrate/mammalian studies, including in zebrafish and mice, due to high degree of conservation of genetic pathways and reduced genetic complexity (*Bodmer and Frasch, 2010*), e.g. the first cardiogenic transcription factor *Nkx2-5/tinman*, discovered in *Drosophila* (*Azpiazu and Frasch, 1993*; *Bodmer, 1993*), marked the beginning of a molecular-genetic understanding of cardiac specification (*Bodmer, 1995*; *Cripps and Olson, 2002*; *Benson et al., 1999*; *Cordes and Srivastava, 2009*; *Qian and Srivastava, 2013*; *Kathiriya et al., 2015*; *Fahed et al., 2013*).

For this study, we combined whole-genome sequencing (WGS), iPSC technology and model system validation with a family-based approach to identify and characterize novel HLHS-associated candidate genes and postulate potential mechanisms involved. This approach led us to identify LRP2 as a major regulator of cardiac cardiomyocyte proliferation of hiPSCs and heart development and maturation in both *Drosophila* and zebrafish. Consistent with our model system findings, burden analysis revealed that rare and predicted deleterious missense *LRP2* variants were enriched in HLHS patients as compared to healthy controls. Finally, we found evidence consistent with *LRP2* regulating cardiac proliferation and differentiation by potentially modulating growth-associated WNT, SHH, and TP53 pathways. Importantly, our integrated multidisciplinary high-throughput approach establishes a scalable and synergistic gene discovery platform to investigate potential oligogenic participants in genetically complex forms of human heart diseases.

## Results

### Transcriptome and cell cycle activity are altered in HLHS patient-derived iPSCs and CMs

This study analyzed a family comprised of unrelated parents and their three offspring ('5H' family; *Figure 1A*). The male proband (II.3) was diagnosed with non-syndromic HLHS by physical examination and echocardiography, which demonstrated aortic and mitral valve atresia, virtual absence of

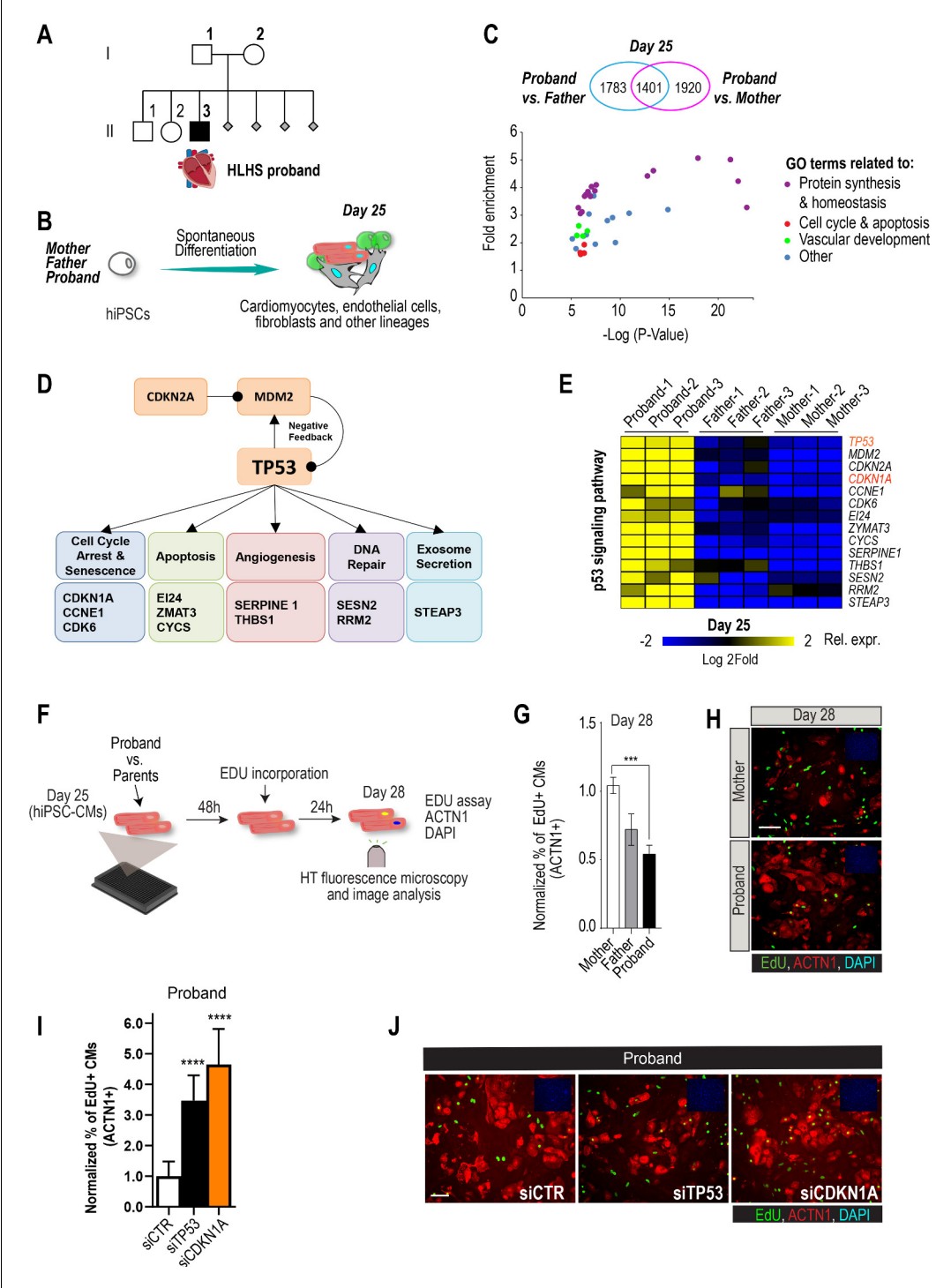

**Figure 1.** Family-based iPSC characterization for HLHS. (**A**) Pedigree of family 5H: proband with HLHS (black symbol), relatives without CHD (white symbols), miscarriages (gray diamonds). (**B**) Schematic for family-based iPSC production and characterization. (**C**) Whole-genome RNA sequencing identified 1401 concordantly DETs between proband and parents. (**D**) KEGG pathway analysis shows enrichment of DETs in TP53 pathway. (**E**) Heatmap of p53 signaling pathway-associated genes in probands vs parents. (**F**) Schematic describing EdU-incorporation assay in hiPSC-CMs. 5000 cells/well were plated in 384 well plates. After 48 hr EdU was added to the media and left incorporate for 24 hr. Cells were then fixed and stained (**G**) Graph representing quantification of EdU+ cardiomyocytes in HLHS 5H family-derived iPSC-CMs. ***p<0.001 one-way ANOVA. (**H**) Representative images of iPSC-CMs derived from mother (Top) and proband (Bottom), stained for EdU, ACTN1 and DAPI. Scale bar: 50 μm. (**I**) Quantification of EdU-incorporation assay in 5H proband iPSC-CM upon KD of TP53 or CDKN1A. ****p<0.0001, one-way ANOVA. (**J**) Representative images of 5H proband iPSC-CM stained for EdU and ACTN1 upon KD of TP53 or CDKN1A at day 28. Scale bar: 50 μm.

*Figure 1 continued on next page*

*Figure 1 continued*

The online version of this article includes the following figure supplement(s) for figure 1:

**Figure supplement 1.** Cell cycle activity is altered in HLHS patient-derived iPSCs and CMs.

**Figure supplement 2.** Cell cycle activity is altered in HLHS patient-derived iPSCs and CMs.

the left ventricular cavity, and severe aortic arch hypoplasia. He was born prematurely at 29 weeks gestation and underwent staged surgical palliation at 2 and 11 months of age. Conversion to a fenestrated Fontan circulation at 3 years of age failed owing to systolic and diastolic heart failure, necessitating early take-down. The patient subsequently died of multi-organ system failure. Echocardiography revealed structurally and functionally normal hearts in the proband's mother (I.2), father (I.1) and siblings (II.1 and II.2). Maternal history is notable for four miscarriages.

Patient-derived iPSCs are a valuable tool to investigate heart defects, such as those observed in HLHS (*Theis et al., 2015a*; *Hrstka et al., 2017*). In this study, iPSCs from the mother (I.2), father (I.1) and HLHS proband (II.3) were generated (*Takahashi and Yamanaka, 2006*) to investigate differences in transcriptional profiles potentially associated with HLHS. Cells from the proband-parent trio were differentiated to day 25 (d25), using a cardiogenic differentiation protocol and processed for subsequent RNA sequencing (*Figure 1B*). In this in vitro cellular context, bioinformatic analysis revealed 5104 differentially expressed transcripts (DETs) in d25 differentiated samples between proband vs. mother/father (*Supplementary file 1*, Benjamini-corrected $p<0.001$). We found that 1,401 DETs were concordantly differentially expressed between proband and both parents (*Figure 1C*, *Figure 1—figure supplement 1*, *Supplementary files 1*, *2*). Consistent with previous observations in HLHS fetuses (*Gaber et al., 2013*), KEGG analysis revealed TP53 pathway enrichment (*Figure 1D*), including cell cycle inhibition (*Figure 1E* and *Figure 1—figure supplement 1*), consistent with cell proliferation being affected in proband cells.

To begin exploring this hypothesis, we measured cell cycle activity in proband and parent hiPSC-derived cardiomyocytes (hiPSC-CMs) using an EdU-incorporation assay (*Figure 1F*). Indeed, proband hiPSC-CMs exhibited reduced percentage of EdU-positive cells as compared to parents (*Figure 1G, H*). To further evaluate whether a potentially reduced proliferative activity is a more general phenotypic hallmark of HLHS cells, we evaluated the proliferative status of two additional HLHS family trios that were available to us from the HLHS cohort at Mayo Clinic ('75H', '151H'). Consistent with our findings with 5H family-trio cells (*Figure 1G,H*), the proband cells of families 75H and 151H also exhibited significant reduction of proliferative activity as compared to the parents using the EdU-incorporation assay (*Figure 1—figure supplement 2*). Given the upregulation of potent cell cycle inhibitors *TP53* or *CDKN1A* in 5H proband cells (*Figure 1E*), we tested whether impaired proliferation could involve the observed elevated *TP53* and/or *CDKN1A* mRNA levels. Indeed, siRNA-mediated knockdown (KD) of TP53 and CDKN1A in proband hiPSC-CMs significantly increased EdU incorporation as compared to siControl (*Figure 1I,J*). These findings are consistent with a CM proliferation defect observed in both HLHS fetuses (*Gaber et al., 2013*) and a HLHS mouse model (*Liu et al., 2017*).

## Family-based WGS, variant filtering, and transcriptional profiling identified 10 candidates

Array comparative genome hybridization ruled out a chromosomal deletion or duplication in the proband. WGS was carried out on genomic DNA samples from all five family members, based on 101 base paired-end reads that passed quality control standards; 92% of the reads mapped to the genome. After marking and filtering out duplicate reads, over 99% of the hg19 human reference genome had coverage. The average depth across the genome was 36X and an average of 91% of the gene body regions (exons, introns, and 5' and 3' untranslated regions) demonstrated a minimal read depth of 20 reads. WGS was performed to identify potentially pathogenic coding or regulatory single nucleotide variants (SNVs) or insertion/deletions (INDELs). First, we ruled out pathogenic variants within 42 genes comprising a CHD genetic testing panel (Invitae, San Francisco, CA). To identify novel HLHS candidate genes, WGS of the family quintet was filtered for rare de novo, recessive and loss-of-function variants with predicted impact on protein structure or expression, yielding 114 variants in 61 genes (*Figure 2*, *Figure 2—figure supplement 1,*, *Supplementary file 3*). We next

prioritized genes most likely to drive downstream pathways of dysregulated cardiogenesis in the HLHS proband by cross-referencing these candidate genes with 3,816 DETs identified in undifferentiated iPSC at d0 (*Supplementary file 4*) and 5,104 DETs identified at d25 differentiated cell lineages (*Supplementary file 1*). Ten genes harboring compound heterozygous (7), hemizygous (2), or homozygous (1) recessive variants (*Table 1*), absent in the unaffected siblings, were found to be differentially expressed within the HLHS proband's iPSCs at d0 and d25: *HSPG2, APOB, LRP2, PRTG, SLC9A1, SDHD, JPT1, ELF4, HS6ST2,* and *SIK1* (*Figure 2A*). qPCR confirmed reduced expression of these genes in proband in d25 iPSC-CM, compared to the parental cells (*Figure 2—figure supplement 2*). In order to explore if and how these genes could affect cardiac differentiation and/or function, alone or in combination, we employed an integrated gene discovery platform using multiple genetic model systems (see below). We consider this approach an efficient first pass evaluation of the potential roles of these genes in the heart; roles that need to be further substantiated by validation of patient-specific variants, also in a combinatorial fashion, based on the oligogenic hypothesis of CHDs.

**Table 1.** Recessive Variants Identified in 10 Candidate Genes.

| Gene | Mode of inheritance | Functional impact | Transcript variant | Protein variant | Inheritance | Genotype in brother (II.1) | Genotype in sister (II.2) | gnomAD* MAF (%) | dbSNP ID |
|------|------|------|------|------|------|------|------|------|------|
| HSPG2 | Cmpd Het | missense | c.2074G > A; c.2077G > A | p.V692M; p.V693M | Maternal | WT | WT | 0.288 | 143669458 |
| | | missense | c.326G > A | p.R109Q | Paternal | Het | Het | 0 | 773796176 |
| | | promoter | c.-227C > A | | Paternal | WT | WT | 1.392 | 566166086 |
| SLC9A1 | Cmpd Het | promoter | c.-906T > C | | Paternal | Het | Het | 1.227 | 114101904 |
| | | promoter | c.-947T > G | | Maternal | WT | WT | 27.175 | 11588974 |
| | | promoter | c.-1085A > G | | Paternal | Het | Het | 0.841 | 116299278 |
| | | ENCODE TFBS | c.-1138C > T | | Paternal | Het | Het | 0.93 | 75089536 |
| | | promoter | c.-1311G > A | | Paternal | Het | Het | 0.93 | 77414471 |
| APOB | Cmpd Het | missense | c.13441G > A | p.A4481T | Maternal | WT | Het | 2.475 | 1801695 |
| | | missense | c.751G > A | p.A251T | Paternal | Het | WT | 0.071 | 61741625 |
| LRP2 | Cmpd Het | missense | c.9613A > G | p.N3205D | Maternal | WT | WT | 0.407 | 35734447 |
| | | missense | c.170C > T | p.A57V | Paternal | WT | WT | 0.032 | 115350461 |
| SDHD | Cmpd Het | promoter | c.-815G > C; c.129+547C > G | | Maternal | Het | WT | 0.573 | 117661257 |
| | | ENCODE TFBS | c.-205G > A; c.66C > T | p.A22A | Paternal | WT | WT | 0.241 | 61734353 |
| | | missense | c.34G > A; c.-173C > T | p.G12S | Maternal | Het | WT | 0.729 | 34677591 |
| PRTG | Cmpd Het | microRNA Binding Site | c.*3501T > G | | Paternal | Het | WT | 0.739 | 77181316 |
| | | microRNA Binding Site | c.*2678A > G | | Maternal | WT | Het | 0.019 | 756136447 |
| HN1 | Cmpd Het | ENCODE TFBS | c.56+617C > T; c.-903C > T; c.-178+617C > T; c.-590C > T | | Maternal | Het | WT | 3.764 | 117213586 |
| | | promoter | c.-1748A > C; c.-719A > C; c.-486A > C | | Paternal | WT | Het | 0.816 | 73995795 |
| SIK1 | Hom Rec | missense | c.2087C > T | p.P696L | Maternal and Paternal | WT | Het | | 1256991707 |
| ELF4 | X-Linked | missense | c.1144G > A | p.V382I | Maternal | WT | Het | 0.025 | 148953158 |
| HS6ST2 | X-Linked | missense | c.948–40041G > A; c.1046G > A | p.R349Q | Maternal | WT | Het | 0.146 | 201239951 |

Cmpd Het, compound heterozygous; Het, heterozygous; Hom Rec, homozygous recessive; MAF, minor allele frequency; WT, wild-type.

*At study initiation the ESP database was used to set the 3% allele frequency filter. Updated frequencies are shown based on the newer gnomAD database curation which would now eliminate *SLC9A1* and *HN1* as candidate genes.

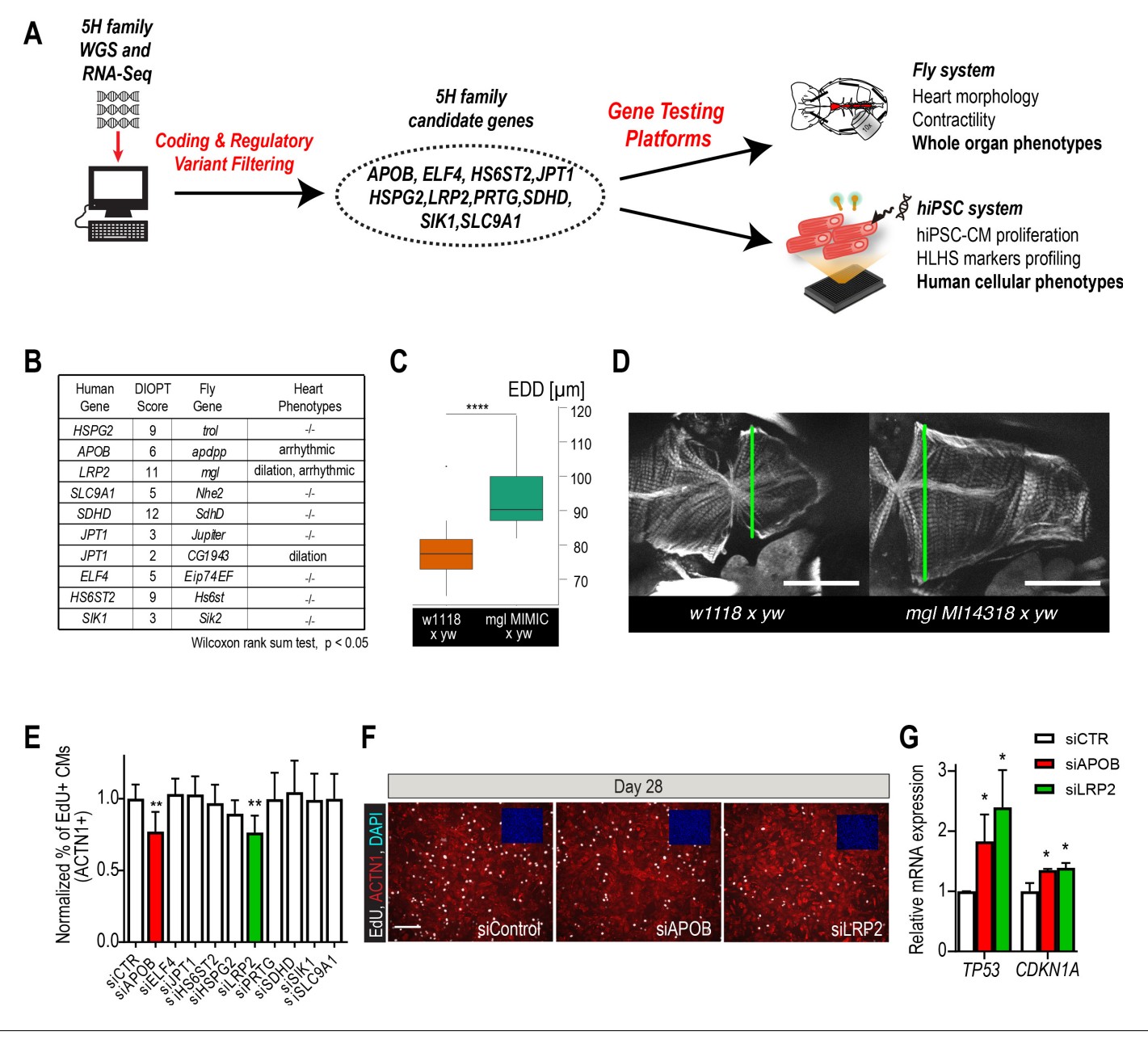

**Figure 2.** Whole-genome and RNA sequencing identify HLHS candidate genes. (**A**) An iterative, family-based variant filtering approach based on rarity, functional impact, and mode of inheritance and RNA sequencing data were used to filter for transcriptional differences yielding 10 candidate genes. Candidate genes were further tested in hiPSC-CM and in vivo model. (**B**) Human candidate genes and corresponding *Drosophila* ortholog as determined by DIOPT score (*confidence score: number of databases reporting orthology). Listed are heart phenotypes upon gene candidate KD. (**C**, **D**) Example of fly hearts heterozygous for *LRP2/mgl* show increased end-diastolic diameters (EDD, measured at green line in **D**). Wilcoxon rank sum test: ***p<0.001. (**E**) Graph representing EdU-incorporation assay results of candidate gene KD in hiPSC-CM. KD of APOB (red bar) or LRP2 (green bar) reduced EdU incorporation. **p<0.01 one-way ANOVA. (**F**) Representative images of hiPSC-CMs stained for EdU, ACTN1 and DAPI. Scale bar: 50 μm. (**G**) qPCR results of TP53 and CDKN1A in hiPSC-CM upon KD of APOB or LRP2. *p<0.05 one-way ANOVA.

The online version of this article includes the following figure supplement(s) for figure 2:

**Figure supplement 1.** Identification of HLHS candidate genes from whole-genome and RNA sequencing.

**Figure supplement 2.** Cell cycle activity is altered in HLHS patient-derived iPSCs and CMs.

**Figure supplement 3.** Phenotypic assessment of HLHS candidate genes in *Drosophila* adult hearts.

**Figure supplement 4.** *LRP2* and *APOB* KD reduces total nuclei and affect cell cycle in hiPSC-CMs.

## Knockdown of candidate gene orthologs in *Drosophila* heart

In order to determine whether these variants occurred within genes that could be important for cardiac differentiation in vivo, we took advantage of our established *Drosophila* heart development model and functional analysis tools (*Figure 2—figure supplement 3*; *Ocorr et al., 2014*). We hypothesized that genes critical for the *Drosophila* heart have conserved roles, also in humans, as previously observed (*Bodmer, 1995*; *Cripps and Olson, 2002*; *Qian et al., 2011*). Predicted by DIOPT database (*Hu et al., 2011*) to have orthologs in *Drosophila* (*Figure 2B*), we analyzed nine genes using heart-specific RNAi-KD. By in vivo heart structure and function analysis (*Fink et al., 2009*; *Ocorr et al., 2014*), we found that KD of *LRP2* (*mgl*) and *JPT1* (*CG1943*) caused dilated heart phenotypes, while KD of *APOB* (*apolpp*), a circulating lipoprotein ligand, and again *LRP2* (*mgl*), resulted in arrhythmias (*Figure 2B–D*, *Figure 2—figure supplement 3*; *Videos 1–3*), suggesting developmental defects of cardiac structure and function.

Since HLHS is likely oligogenic (*Blue et al., 2017*; *Gelb and Chung, 2014*), functional requirements for some genes involved in HLHS might only become apparent in combination with variants in other cardiac-relevant genes. To test this, we examined the nine candidates in the heterozygous background for *tinman/NKX2-5*, which in humans is well-known to contribute to a variety of CHD/HLHS manifestations (*Elliott et al., 2003*; *Hrstka et al., 2017*; *Benson, 2010*; *Kobayashi et al., 2014*). In this in vivo context, heart-specific KD of two out of nine genes, *HSPG2/Perle*can (*trol*), involved in extracellular matrix assembly (*Sasse et al., 2008*), and Succinate dehydrogenase subunit D *SDHD* (*SdhD*) exhibited a constricted phenotype (*Figure 2—figure supplement 3G,H*). These findings demonstrate that our bioinformatic candidate gene prioritization identified several conserved candidates as cardiac relevant, but further validation is necessary to begin to link them in a causal fashion to HLHS.

## *LRP2* and *APOB* regulate proliferation in human iPSC-derived cardiomyocytes

Decreased proliferation of left ventricular cardiomyocytes is emerging as a phenotypic hallmark of HLHS (*Liu et al., 2017*; *Gaber et al., 2013*) (see also *Figure 1G,H* and *Figure 1—figure supplement 2*), suggesting that cell cycle impairment may be an important contributing factor. Thus, we asked whether siRNA-mediated KD of the prioritized 10 candidate genes from the 5H family trio (*Figure 2A*) affects proliferation of healthy, normal hiPSC-CM (*Cunningham et al., 2017*). Remarkably, two of the genes causing cardiac abnormalities when knocked down in *Drosophila* (*Figure 2B–D*), *LRP2* and *APOB*, also caused a marked reduction of EdU+ hiPSC-CMs (ACTN1+) and overall hiPSC cell numbers (*Figure 2E,F* and *Figure 2—figure supplement 4A,B*). Notably, we also observed an upregulation of cell cycle inhibitors and apoptosis genes (*Figure 2—figure supplement 4C*), including TP53 and CDKN1A (*Figure 2G*), as well as a downregulation of cell cycle genes (*Figure 2—figure supplement 4B,C*). Collectively, these data identify *LRP2* and *APOB* as modulators of cell cycle and apoptosis in hiPSC-CMs, however, further validation is necessary to link them to contributing to the developmental cardiac impairment in HLHS patients.

## Rare variant analysis in HLHS cohort reveals enrichment in *LRP2*

In order to explore disease relevance of candidate genes functionally validated in both systems, we asked whether the frequency of rare and predicted-damaging variants in *LRP2* and *APOB* would be higher in a cohort of 130 HLHS cases compared to 861 control individuals. Remarkably, HLHS patients had a ~ 3 fold increase in the frequency of rare, predicted-damaging *LRP2* missense variants compared to healthy controls (10% versus 3.4%; p=0.0008) (*Figure 3A*; *Supplementary file 6*). Among the 13 patients who carried a *LRP2* variant (*Figure 3A,B*), three shared the same

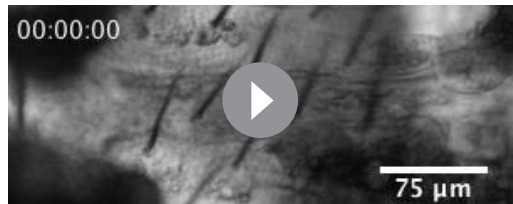

**Video 1.** Dissected adult fly heart showing rhythmic beating pattern. Representative heart movies of dissected adult females showing arrhythmic beating pattern in APOB-RNAi (*Video 2*) and *LRP2*-RNAi (*Video 3*) compared to control hearts (*Video 1*). All movies are imaged at 140 frames/sec.
https://elifesciences.org/articles/59554#video1

**Video 2.** APOB-RNAi causes arrhythmia in dissected adult fly hearts.

https://elifesciences.org/articles/59554#video2

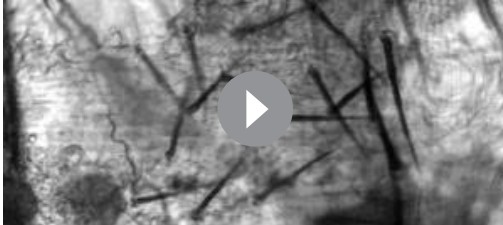

**Video 3.** LRP2-RNAi causes arrhythmia in dissected adult fly hearts.

https://elifesciences.org/articles/59554#video3

predicted-damaging variant (N3205D) with the 5H proband (*Figure 3B*, *Supplementary file 7*). Of note, 13 of the 130 HLHS cases (including the index family proband) possessed <80% of ancestral Caucasian alleles, while all controls possessed ≥80%. Four of the 13 cases had rare, predicted-damaging missense variants in *LRP2*. However, all assessed variants were required to be rare in all racial populations. To eliminate the potentially confounding variable of race a Caucasian-only sub-analysis was performed, resulting in a less significant p-value for rare, predicted-damaging missense variants (7.7% versus 3.4%; p=0.05). However, removal of the predicted-damaging (CADD) restriction on rare *LRP2* variants among Caucasians revealed significant enrichment in cases (p=0.0035), most notably in missense and intronic variants (p=0.0178 and 0.0082, respectively, *Supplementary file 8*). Population-based allele frequencies, CADD scores, and location of variants within functional protein binding domains, active histone marks, or transcription factor binding sites was not different between cases and controls.

In a next step, we sought to determine whether LRP2 levels might be affected in probands with rare, predicted-damaging variants in LRP2 coding sequence. We profiled *LRP2* transcripts levels in patient-derived iPSCs of the 5H family as well as another family, 49H, both harboring heterozygous variants with a CADD score above 24 (*Figure 3B*), inherited from one of the parents. Interestingly, qPCR results showed that LRP2 mRNA levels were lower in the probands of both families, as well as in the parent carrying the variant (CP), compared to non-variant carrying parent (NCP) (*Figure 3C, D*). We do not know why the protein coding variants (see *Table 1*) are associated with reduced RNA levels in patient-derived iPSC-CMs, which we speculate may be due to reduced stability of the variant *LRP2* mRNA, or altered LRP2 function could feed back to reduce expression. This corroborates the idea that these LRP2 variants (N3205D and A3344T) may be causing a genetic loss-of-*LRP2*-function in the 5H and 49H families. However, there are likely other contributing factors besides the presence of the *LRP2* variants, since echocardiography excluded CHD in carrier parents.

## Zebrafish *LRP2* loss-of-function results in a hypoplastic ventricular phenotype

In order to evaluate the role of LRP2 during heart development in a vertebrate model, we injected a morpholino as well as sgRNA/CRISPR directed against *LRP2* (*lrp2a*) in zebrafish embryos and evaluated the effect on heart morphology and function at 72 hpf. Overall body morphology was similar for morphant and F0 CRISPR edited larva at 72 hpf, compared to controls (*Figure 3—figure supplement 1A–C*). Hearts from larvae with reduced *lrp2a* function displayed a hypoplastic phenotype with decreased CM number (*Figure 3E–G*) and dose-dependent reductions in ventricular chamber dimensions in morphants (*Figure 3—figure supplement 1D,E*). Loss-of-*lrp2a*-function also compromised ventricular contractility and caused bradycardia in both morphants and CRISPR-edited larvae (*Figure 3—figure supplement 1F,G*). Collectively, our data suggest that *LRP2* plays a crucial role during heart development by regulating cardiomyocyte generation most prominently in the ventricular chamber.

## Potential regulatory network of validated gene candidates

In order to delineate how the candidate genes testing positive in our validation systems, *APOB*, *HS6ST2*, *HSPG2*, *JPT1*, *LRP2*, might affect signaling homeostasis, we assembled a gene network containing these five genes and their first neighbors (as genetic and protein-protein interactions,

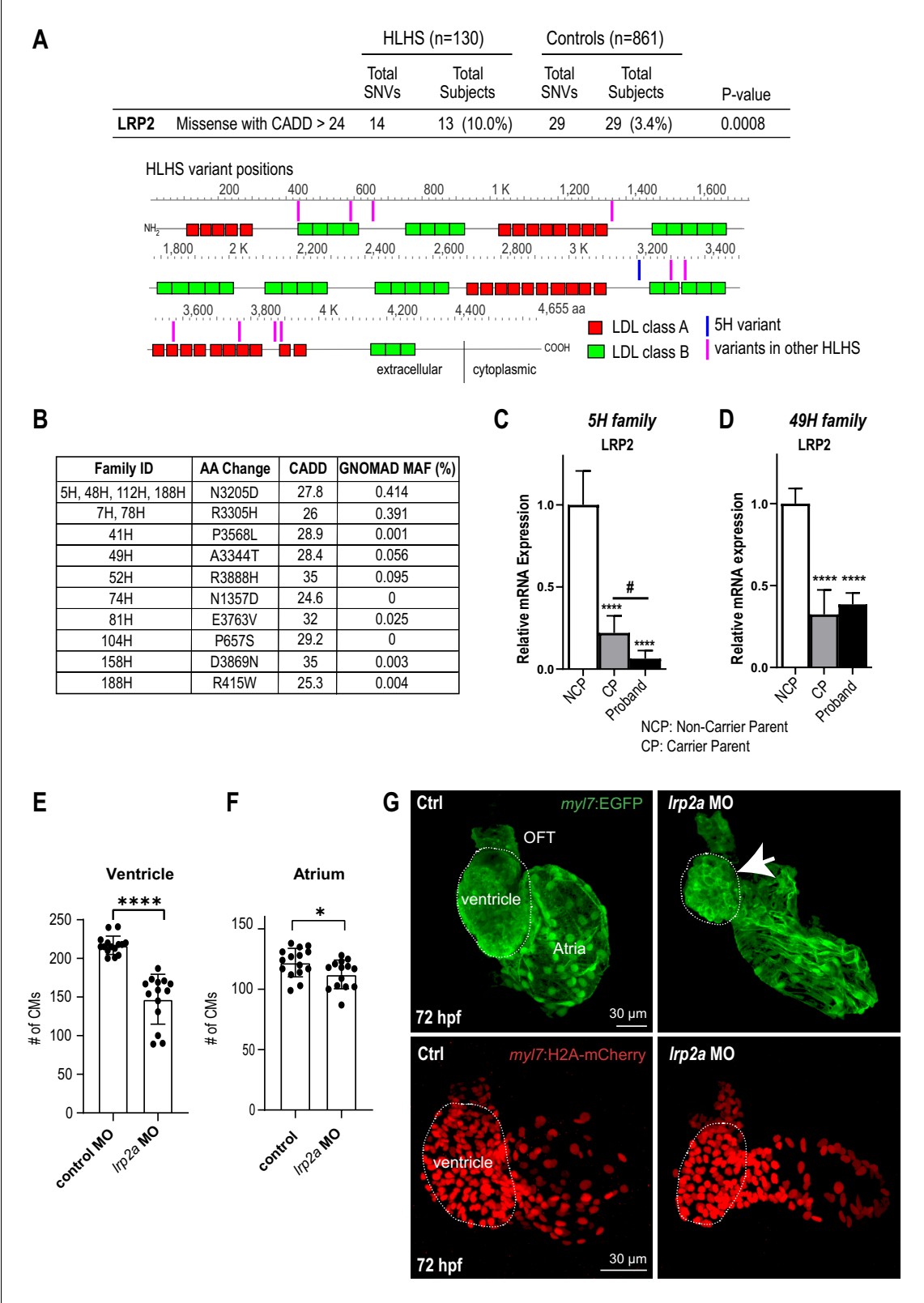

**Figure 3.** Identification of LRP2 as potential HLHS candidate gene. (**A**) Cohort-wide analysis of LRP2 variants shows significant enrichment for SNVs in HLHS patients compared to control populations. Variants (blue/magenta) are found throughout LRP2 protein. (**B**) Table listing the HLHS families carrying LRP2 variants. (**C,D**) qPCR of LRP2 in 5H family (**C**) and in 49H family (**D**) showing LRP2 downregulation in carrier parent and proband compared to the non-carrier parent. ****p<0.0001 one-way ANOVA #p<0.05 one-way ANOVA. (**E**) Cardiomyocyte count in zebrafish morphants at 72 hpf were

*Figure 3 continued on next page*

*Figure 3 continued*

significantly reduced in the ventricle. (F) Atrial cardiomyocyte number was also reduced in morphants but to a lesser extent than in ventricles. *p<0.05; ****p<0.0001 unpaired two-tail Student t-test. (G, top panel) embryonic fish hearts were visualized by EGFP expression in the *myl7:EGFP* transgenic background (green) at 72 hpf. *lrp2a* morphant hearts were dysmorphic and much smaller (arrow) compared to controls. (G, lower panel) *myl7:H2A-mCherry* transgenic background identifies cardiomyocyte nuclei used for quantifying cardiomyocytes during development in E and F. Dotted traces outline the ventricles in G. Scale bars: 30 μm.

The online version of this article includes the following figure supplement(s) for figure 3:

**Figure supplement 1.** *lrp2a* KD and CRISPR causes reduced contractility and bradycardia in zebrafish larva.

BioGRID) (*Figure 4A*; *Supplementary file 5*). Strikingly, in addition to TP53 pathway misregulation (*Figure 1D–E*, *Figure 2G* and *Figure 2—figure supplement 4*), all five genes were connected to WNT and SHH signaling cascades, both key regulators of cardiac differentiation and proliferation (*Briggs et al., 2016*; *Gessert and Kühl, 2010*; *Figure 4A*). Interestingly, RNA-seq analysis of the proband cells is consistent with this network prediction: the negative regulator of SHH pathway, *PTCH1*, was upregulated, while agonists of WNT signaling pathway, *WNT1/3a/8a/10b* and *FZD10* (*Dawson et al., 2013*), were downregulated, compared to parental cells. However, genetic interaction studies in our model systems are required in order to substantiate a link between these two pathways and LRP2 (and APOB) in CM proliferation.

As a first approach, we examined whether LRP2 could regulate WNT- and/or SHH-associated genes, using healthy control hiPSC-CM. Indeed, KD of LRP2 led to reduced *FZD10* and increased *PTCH1* RNA levels (*Figure 4B,C*), although WNT1/3a/8/10a were not affected. Next, we used the WNT agonist BIO (*Tseng et al., 2006*) and siRNA against PTCH1 (*Kawagishi et al., 2018*) in the presence of LRP2 siRNA. We found that LRP2 KD significantly reduced both BIO- and siPTCH1-induced proliferation in hiPSC-CMs (*Figure 4D–G*), suggesting that LRP2 is required for both WNT- and SHH-regulated CM proliferation. Further experiments are required to substantiate a link between LRP2 and SHH/WNT signaling in heart growth and differentiation.

## Discussion

### Integrated multidisciplinary disease gene discovery platform

Unraveling the patient-specific molecular-genetic etiology of HLHS pathogenesis will improve our ability (1) to provide individual diagnostics to families and (2) to develop novel approaches to treat or (3) prevent the disease. As an important first step toward these goals, our integrated multidisciplinary approach is able to identify variants and gene functions, emanating from WGS of family trios, that are relevant for cardiac development and differentiation. Variants in these genes are proposed candidates to potentially contribute to disease etiology. As an example of our heart disease gene discovery platform, we identified *LRP2* as a novel candidate CHD gene with rare variants that are enriched in HLHS patients, thus generating hypotheses for further studies.

In this study, we used the powerful combination of high-throughput DNA/RNA patient sequencing coupled with high-throughput functional screening in model systems enabling to probe gene function (alone or in combination) on a wide array of cellular processes that are deployed during heart formation. For validation in model systems, we have established an integrated multi-site and multidisciplinary pipeline that systematically evaluates the functional role of genes presenting rare and deleterious variants in HLHS patients in hiPSC, *Drosophila* and zebrafish heart models. As a main objective – identify and functionally evaluate genes potentially associated with CHD/HLHS – our study highlights rare, predicted-damaging *LRP2* missense variants as 3-fold enriched in 130 HLHS patients compared to 861 controls. Validation of this gene in hiPSCs, *Drosophila* and zebrafish heart models demonstrated a requirement in cardiac proliferation and differentiation, and notably, systemic KD in zebrafish resulted in ventricular cardiac but not obvious skeletal muscle defects (see also *Figure 3—figure supplement 1*). Mutations in *LRP2* have been previously associated with left ventricular non-compaction (LVNC) as well as other congenital heart defects in mouse (*Baardman et al., 2016*) and in Donnai-Barrow Syndrome in humans (*Baardman et al., 2016*; *Kantarci et al., 2008*). In fact, LVNC is often accompanied by other CHDs (*Stähli et al., 2013*). However, *LRP2* has not previously been linked to HLHS within curated bioinformatic networks. As many

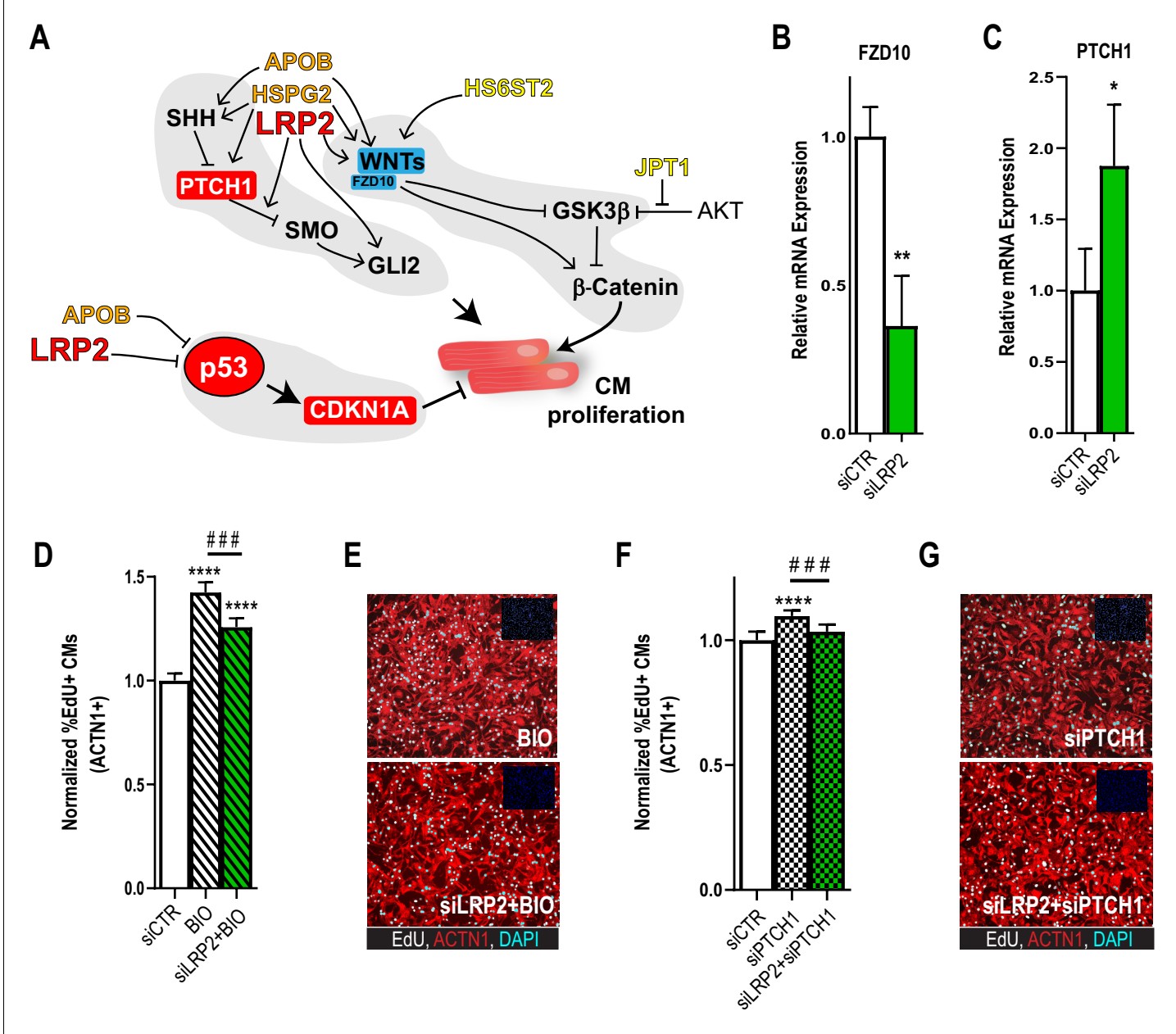

**Figure 4.** Potential role for SHH, WNT and LRP2 in HLHS. (**A**) A gene network integrating family-centric HLHS candidate genes with heart development. ORANGE – genes with cardiac phenotypes in iPSC/*Drosophila* assays. YELLOW – other candidate genes with *Drosophila* phenotypes. RED – Genes up-regulated in proband iPSC-CMs vs. parents. BLUE – Genes downregulated downregulated in proband iPSCs vs. parents. (**B,C**) qPCR for FZD10, a WNT-pathway-associated gene, (**B**) and for PTCH1, a SHH pathway-associated gene (**C**) upon LRP2 KD in hiPSC-CM. **$p<0.01$, *$p<0.05$ Student's t-test. (**D**) Quantification of EdU- incorporation assay in hiPSC-CM upon LRP2 KD in combination with BIO, a WNT inhibitor. ***or ###$p<0.001$, ****$p<0.0001$, one-way ANOVA. (**E**) Representative images of hiPSC-CM stained for EdU and ACTN1. Scale bars: 50 μm. (**F**) Quantification of EdU-incorporation assay in hiPSC-CM upon LRP2 KD in combination with PTCH1 KD, a SHH-associated gene. ***$p<0.001$, ****$p<0.0001$, one-way ANOVA. (**G**) Representative images of hiPSC-CM stained for EdU and ACTN1. Scale bars: 50 μm.

top CHD gene candidates, such as *Nkx2-5* and *Notch*, *LRP2* could also be involved in the etiology of relatively different CHDs, such as LVNC and HLHS. This may again be due to the fact that CHDs in general and HLHS in particular are likely oligogenic and share many common factors.

One pre-requisite to reduce the knowledge gap between patient genomes and clinical pheno-types is to establish reliable/quantifiable phenotypic links between CHD/HLHS candidate genes and their role during normal cardiac development. Given that large-scale genomic studies to identify

CHD-associated genes can each generate hundreds of candidates, we have demonstrated here that our cardiac phenotypical platform is able to perform high-throughput functional screening to accommodate rapid testing of a large number of genes. Although overall heart structure in flies differs from that in vertebrates, the fundamental mechanisms of heart development and function are remarkably conserved, including a common transcriptional regulatory network (*Bodmer, 1995*; *Cripps and Olson, 2002*), a shared protein composition (*Cammarato et al., 2011*), as well as electrical and metabolic properties (*Ocorr et al., 2014*; *Ocorr et al., 2007*; *Diop and Bodmer, 2015*). This 'convergent biology' approach identified *LRP2* as a novel HLHS candidate gene in both the in vitro and in vivo cardiac model systems, although a definite link must await further study. Importantly, variants in *LRP2* were not only found to be enriched in a cohort of 130 HLHS family trios, but also produced a ventricular hypoplastic phenotype in zebrafish embryos upon loss-of-*lrp2a*-function. Therefore, for further mechanistic understanding of complex CHD characterized by oligogenic etiologies this triple model system testing approach enables assessment of gene function combinatorically and in sensitized genetic backgrounds (e.g. *tinman*/NKX2-5; see *Figure 2—figure supplement 3*). Furthermore, the various LRP2 coding variants can now be tested in hiPSC-CM, fish and fly models using CRISPR/Cas technologies, and evaluate whether the specific variant mimics the KD phenotype. Patient-derived proliferation-impaired hiPSC-CM harboring LRP2 variants could be 'corrected' to rescue the defect in a variant/patient-specific manner. Our platform could serve as a general strategy for a first evaluation of candidate genes prioritized from genomic and bioinformatic analysis, before more effort- and time-consuming follow-up studies are undertaken (e.g. *Vissers et al., 2020*).

## A hypothetical pathogenic role for SHH, WNT, p53 and cell proliferation in HLHS

Our current understanding of the molecular-genetic causes of HLHS is very limited, despite clear genetic origins of disease (*Yagi et al., 2018*). Past research on HLHS has yielded very few high-confidence gene candidates that may contribute to HLHS, for example, *NOTCH1*, *NKX2-5* and *MYH6* have been implicated with HLHS (*Elliott et al., 2003*; *Theis et al., 2015a*; *Theis et al., 2015b*), but they are also associated with other CHDs.

Heart development is a complex process that involves the interaction of many pathways and tissues, and a large number of genes have been implicated in various types of CHDs (*Pierpont et al., 2018*). The postulated oligogenic nature of HLHS is likely the result of an unfavorable combination of disease genes, and such a combination of alleles, in turn, could affect several, successive steps of heart development. This makes it extremely difficult to model the disease by single gene mutations. Current hypotheses of the etiology of HLHS include changes in cell cycle progression of myocytes, as well as altered blood flow ('no flow – no grow') as a consequence of defects of valves or the outflow tract (*Grossfeld et al., 2019*; *Saraf et al., 2019*).

Interestingly, the only mouse HLHS model to date, a digenic mutant for *Sap130* and *pcdha9* (*Liu et al., 2017*), has a penetrance of less than 30%, indicating a profound role for subtle differences between genetic backgrounds. This mouse model suggests a separate mechanism with pcdha9 affecting aortic growth, whereas *Sap130* can exert a more severe HLHS-like phenotype, which might reflect a modular etiology of HLHS that separates valve and ventricular defects.

The gene network analysis that we have conducted points to the possibility that several of the prioritized candidate genes identified in the index 5H patient may have a modulatory impact on proliferation and differentiation, potentially via WNT/SHH-associated pathways (*Briggs et al., 2016*). There is evidence that the three of the candidates with fly heart phenotypes – *Trol/HSPG2*, *Mgl/LRP2* and *Apolpp/APOB* – can alter WNT and SHH signaling (*Christ et al., 2015*; *Datta et al., 2006*), but future studies – for example genetic interaction experiments in our model systems – are needed to support their involvement. We hypothesize that a collective of likely hypomorphic genetic variants affects heart development leading to HLHS. Impaired ventricular growth could in addition be caused by changes in the p53 pathway, and our analysis of iPSC-derived cardiomyocytes suggests that p53 indeed depends on LRP2 levels. Such a multi-hit model of HLHS caused by subthreshold hypomorphic alleles represents an attractive explanation of many CHDs.

In summary, this integrated multidisciplinary strategy of functional genomics using patient-specific iPSC combined with in vivo and human cellular model systems of functional validation has much promise in generating hypotheses, such as novel genetic pathways and potential polygenic

interactions underlying CHD/HLHS. Evaluating patient-specific, complex polygenic risk factors potentially underlying HLHS will likely establish the groundwork for definitive mechanistic studies of interacting risk factors that contribute to defective cardiac development and adverse outcomes. This scalable approach promises more efficient discovery of novel CHD/HLHS gene candidates and multiple HLHS families can now be multiplexed in future diagnostic and therapeutic studies.

# Materials and methods

## Key resources table

| Reagent type (species) or resource | Designation | Source or reference | Identifiers | Additional information |
|---|---|---|---|---|
| Genetic reagent (*D. melanogaster*) | *Hand*[4.2]-Gal4 | NA | PMID:16467358 | NA |
| Genetic reagent (*D. melanogaster*) | UAS-trol[RNAi] | Vienna *Drosophila* Resource Center (VDRC) | FBst0454629 | v22642 |
| Genetic reagent (*D. melanogaster*) | UAS-CG1943[RNAi] | Vienna *Drosophila* Resource Center (VDRC) | FBst0453803 | v20758 |
| Genetic reagent (*D. melanogaster*) | UAS-apolpp[RNAi] | Vienna *Drosophila* Resource Center (VDRC) | FBst0470481 | v6878 |
| Genetic reagent (*D. melanogaster*) | UAS-Hs6st[RNAi] | Vienna *Drosophila* Resource Center (VDRC) | FBst0464695 | v42658 |
| Genetic reagent (*D. melanogaster*) | UAS-mgl[RNAi] | Vienna *Drosophila* Resource Center (VDRC) | FBst0461660 | v36389 |
| Genetic reagent (*D. melanogaster*) | UAS-Sdhd[RNAi] | Vienna *Drosophila* Resource Center (VDRC) | FBst0456581 | v26776 |
| Genetic reagent (*D. melanogaster*) | UAS-Nhe2[RNAi] | Vienna *Drosophila* Resource Center (VDRC) | FBst0477879 | v106053 |
| Genetic reagent (*D. melanogaster*) | UAS-Jupiter[RNAi] | Vienna *Drosophila* Resource Center (VDRC) | FBst0455704 | v25044 |
| Genetic reagent (*D. melanogaster*) | UAS-Eip74EF[RNAi] | Vienna *Drosophila* Resource Center (VDRC) | FBst0477129 | v105301 |
| Genetic reagent (*D. melanogaster*) | UAS-Sik2[RNAi] | Vienna *Drosophila* Resource Center (VDRC) | FBst0456442 | v26496 |
| Genetic reagent (*D. melanogaster*) | mgl[MI14318] | Bloomington *Drosophila* Stock Center (BDSC) | FBal0302551 | BL-59689 |
| Genetic reagent (*D. melanogaster*) | tin[346] | NA | FBal0035787 | NA |
| Strain, strain background (*D. rerio*) | Oregon AB wild-type | | | A commonly used wild-type strain |
| Strain, strain background (*D. rerio*) | *Tg(myl7:EGFP)*[twu277] | Tsai Lab, National Taiwan University | PMID:12950077 | A transgenic line of zebrafish labeled with heart-specific EGFP fluorescence. |
| Strain, strain background (*D. rerio*) | *Tg(myl7:H2A-mCherry)*[sd12] | Yelon Lab, University of California, San Diego | PMID:24075907 | A transgenic line of zebrafish specifically expressing mCherry in cardiomyocyte nuclei |
| Antibody | mouse monoclonal anti-ACTN1 | Sigma | A7811 | 1:800 |
| Antibody | donkey polyclonal anti-mouse Alexa fluor 568 | Invitrogen | A10037 | 1:1000 |
| Antibody | chicken polyclonal anti-GFP | Aves Labs | GFP-1020 | 1:300 |
| Antibody | rabbit polyclonal abit-mCherry | Rockland | 600–401 P16S | 1:200 |
| Antibody | donkey polyclonal anti-chicken AlexaFluor 488 | Jackson Immuno Research | 703-545-155 | 1:200 |

*Continued on next page*

*Continued*

| Reagent type (species) or resource | Designation | Source or reference | Identifiers | Additional information |
|---|---|---|---|---|
| Antibody | donkey polyclonal anti-rabbit AlexaFluor 568 | Invitrogen | A10042 | 1:200 |
| Other | DAPI (iPSC) 500 mg/mL | Sigma | D9542 | 1:1000 |
| Other | DAPI (Zebrafish) 500 mg/mL | Invitrogen | D1306 | 1:200 |
| Sequence-based reagent | LRP2 siRNA | Entrez Gene ID: 4036 | Dharmacon | On-Target plus, SmartPool |
| Sequence-based reagent | APOB siRNA | Entrez Gene ID: 338 | Dharmacon | On-Target plus, SmartPool |
| Sequence-based reagent | PTCH1 siRNA | Entrez Gene ID: 5727 | Dharmacon | On-Target plus, SmartPool |
| Sequence-based reagent | TP53 siRNA | Entrez Gene ID: 7157 | Dharmacon | On-Target plus, SmartPool |
| Sequence-based reagent | CDKN1A siRNA | Entrez Gene ID: 1026 | Dharmacon | On-Target plus, SmartPool |
| Sequence-based reagent | ELF4 siRNA | Entrez Gene ID: 2000 | Dharmacon | On-Target plus, SmartPool |
| Sequence-based reagent | JPT1 siRNA | Entrez Gene ID: 51155 | Dharmacon | On-Target plus, SmartPool |
| Sequence-based reagent | HS6ST2 siRNA | Entrez Gene ID: 90161 | Dharmacon | On-Target plus, SmartPool |
| Sequence-based reagent | HSPG2 siRNA | Entrez Gene ID: 3339 | Dharmacon | On-Target plus, SmartPool |
| Sequence-based reagent | PRTG siRNA | Entrez Gene ID: 283659 | Dharmacon | On-Target plus, SmartPool |
| Sequence-based reagent | SDHD siRNA | Entrez Gene ID: 6392 | Dharmacon | On-Target plus, SmartPool |
| Sequence-based reagent | SIK1 siRNA | Entrez Gene ID: 150094 | Dharmacon | On-Target plus, SmartPool |
| Sequence-based reagent | SLC9A1 siRNA | Entrez Gene ID: 6548 | Dharmacon | On-Target plus, SmartPool |
| Sequence-based reagent | CDH | Hs00170423_m1 | IDT Integrated DNA technologies, Coralville, IA | characterization of the pluripotent state |
| Sequence-based reagent | DNMT3 | Hs01003405_m1 | IDT Integrated DNA technologies, Coralville, IA | characterization of the pluripotent state |
| Sequence-based reagent | DPPA2 | Hs00414521_g1 | IDT Integrated DNA technologies, Coralville, IA | characterization of the pluripotent state |
| Sequence-based reagent | DPPA5 | Hs00988349_g1 | IDT Integrated DNA technologies, Coralville, IA | characterization of the pluripotent state |
| Sequence-based reagent | ERAS | Hs.PT.45.4849266.g | IDT Integrated DNA technologies, Coralville, IA | characterization of the pluripotent state |
| Sequence-based reagent | GDF3 | Hs00220998_m1 | IDT Integrated DNA technologies, Coralville, IA | characterization of the pluripotent state |
| Sequence-based reagent | OCT-4 | Hs.PT.45.14904310.g | IDT Integrated DNA technologies, Coralville, IA | characterization of the pluripotent state |
| Sequence-based reagent | REXO1 | Hs.PT.45.923095.g | IDT Integrated DNA technologies, Coralville, IA | characterization of the pluripotent state |
| Sequence-based reagent | SALL4 | Hs00360675_m1 | IDT Integrated DNA technologies, Coralville, IA | characterization of the pluripotent state |
| Sequence-based reagent | TDG1 | Hs02339499_g1 | IDT Integrated DNA technologies, Coralville, IA | characterization of the pluripotent state |
| Sequence-based reagent | TERT | Hs99999022_m1 | IDT Integrated DNA technologies, Coralville, IA | characterization of the pluripotent state |

*Continued*

| Reagent type (species) or resource | Designation | Source or reference | Identifiers | Additional information |
|---|---|---|---|---|
| Sequence-based reagent | APOB | Hs.PT.56a.1973344 | IDT Integrated DNA technologies, Coralville, IA | characterization of the pluripotent state |
| Sequence-based reagent | DHCR24 | Hs.PT.56a.4561516 | IDT Integrated DNA technologies, Coralville, IA | expression during guided cardiac differentiation |
| Sequence-based reagent | ELF4 | Hs.PT.56a.25941471 | IDT Integrated DNA technologies, Coralville, IA | expression during guided cardiac differentiation |
| Sequence-based reagent | HN1 | Hs.PT.58.40922463.g | IDT Integrated DNA technologies, Coralville, IA | expression during guided cardiac differentiation |
| Sequence-based reagent | HSPG2 | Hs.PT.56a.18698732 | IDT Integrated DNA technologies, Coralville, IA | expression during guided cardiac differentiation |
| Sequence-based reagent | HS6ST2 | Hs.PT.56a.1354985 | IDT Integrated DNA technologies, Coralville, IA | expression during guided cardiac differentiation |
| Sequence-based reagent | LRP2 | Hs.PT.56a.1584067 | IDT Integrated DNA technologies, Coralville, IA | expression during guided cardiac differentiation |
| Sequence-based reagent | MYLK | Hs.PT.56a.39795491 | IDT Integrated DNA technologies, Coralville, IA | expression during guided cardiac differentiation |
| Sequence-based reagent | PCDH11X | Hs.PT.56a.26531358 | IDT Integrated DNA technologies, Coralville, IA | expression during guided cardiac differentiation |
| Sequence-based reagent | PRTG | custom design | IDT Integrated DNA technologies, Coralville, IA | expression during guided cardiac differentiation |
| Sequence-based reagent | SIK1 | Hs.PT.58.2995158 | IDT Integrated DNA technologies, Coralville, IA | expression during guided cardiac differentiation |
| Sequence-based reagent | SLC9A1 | Hs.PT.58.15072523 | IDT Integrated DNA technologies, Coralville, IA | expression during guided cardiac differentiation |
| Sequence-based reagent | SDHD | Hs.PT.58.40267655.g | IDT Integrated DNA technologies, Coralville, IA | expression during guided cardiac differentiation |
| Sequence-based reagent | GAPDH | Hs.PT.45.8326 | IDT Integrated DNA technologies, Coralville, IA | expression during guided cardiac differentiation |
| Commercial assay or kit | EdU | Click-it Plus EdU Imaging Kit | Life Technologies | |
| Chemical compound, drug | BIO (GSK-3 Inhibitor) | | Sigma | B1686 |
| Software, algorithm | Prism v7 and v8 | | GraphPad Software | |

## Study subjects

Written informed consent was obtained for the index family and an HLHS cohort, under a research protocol approved by the Mayo Clinic Institutional Review Board. Cardiac anatomy was assessed by echocardiography. Candidate genes were identified and prioritized by WGS of genomic DNA and RNA sequencing of patient-specific iPSC and cardiomyocytes. For variant burden analysis, controls were obtained from the Mayo Clinic Center for Individualized Medicine's Biobank. Methods for genomic analyses, RNA Sequencing, iPSC technology, bioinformatics and statistics are described in the Online Appendix. Data are available in NCBI SRA database (see below for SRA Accession IDs).

## Comparative genomic hybridization

To detect aneuploidy, array comparative genomic hybridization was performed using a custom 180K oligonucleotide microarray (Agilent, Santa Clara, CA), with a genome-wide functional resolution of approximately 100 kilobases. Deletions larger than 200 kilobases and duplications larger than 500 kilobases were considered clinically relevant.

## Whole-genome sequencing (WGS) and bioinformatics analyses of index family

Genomic DNA was isolated from peripheral white blood cells or saliva. WGS and variant call annotation were performed utilizing the Mayo Clinic Medical Genome Facility and Bioinformatics Core. Paired-end libraries were prepared using the TruSeq DNA v1 sample prep kit following the manufacturer's protocol (Illumina, San Diego, CA). Each whole-genome library was loaded into four lanes of a flow cell and 101 base pair paired-end sequencing was carried out on Illumina's HiSeq 2000 platform using TruSeq SBS sequencing kit version three and HiSeq data collection version 1.4.8 software. Reads were aligned to the hg19 reference genome using Novoalign version 2.08 (http://novocraft.com) and duplicate reads were marked using Picard (http://picard.sourceforge.net). Local realignment of INDELs and base quality score recalibration were then performed using the Genome Analysis Toolkit version 1.6–9 (GATK) (*McKenna et al., 2010*). SNVs and INDELs were called across all samples simultaneously using GATK's Unified Genotype with variant quality score recalibration (VQSR) (*DePristo et al., 2011*).

Variant call format (VCF) files with SNV and INDEL calls from each family member were uploaded and analyzed using Ingenuity Variant Analysis software (QIAGEN, Redwood City, CA) where variants were functionally annotated and filtered by an iterative process. First, rare, high quality heterozygous variants were selected that (a) had a read depth of at least 10 (b) were not adjacent to a homopolymer exceeding five base pairs (c) were present in <5 whole exome sequencing (WES) datasets collected from 147 individuals not affected with HLHS and (d) were present at a frequency <1% (de novo, loss-of-function, CHD panel genes) or <3% (compound heterozygous, hemizygous or homozygous recessive) in the Exome Variant Server (WES data from 6503 individuals, URL: http://evs.gs.washington.edu/EVS) 1000 Genomes (WGS data from 1092 individuals) (*Abecasis et al., 2012*), and/or Complete Genomics Genome (WGS data from 69 individuals) (*Drmanac et al., 2010*). Second, functional variants were selected, defined as those that impacted a protein sequence, canonical splice site, microRNA coding sequence/binding site, enhancer region, or transcription factor binding site within a promoter validated by ENCODE chromatin immunoprecipitation experiments (*Raney et al., 2014*). Third, using parental and sibling WGS data, rare, functional variants in the proband were then filtered for those that arose de novo or fit homozygous recessive, compound heterozygous, or X-linked recessive modes of inheritance. In addition, any inherited frameshift and start/stop codon variants were retained if they occurred in a gene intolerant to loss-of-function (pLI score > 0.75).

## WGS of an HLHS cohort and unaffected controls

WGS of 130 unrelated individuals with left ventricular hypoplasia (HLH) (80% HLHS, 20%CHD with HLH) was performed utilizing the Mayo Clinic Medical Genome Facility. For the control population, 861 individuals from the Mayo Clinic Center for Individualized Medicine's Biobank repository (*Olson et al., 2013*) were selected based upon absence of personal or family history of CHD and underwent WGS at HudsonAlpha Institute for Biotechnology. Variant call annotation for all 991 individuals was performed by the Mayo Clinic Bioinformatics Core. Whole-genome libraries were prepared for 130 individuals with HLHS and 101 bp or 150 bp paired-end sequencing was performed on either the Illumina HiSeq 2000 (n = 56) or HiSeq 4000 (n = 74), respectively. For the 861 Biobank individuals, whole-genome libraries were prepared, and 150 base pair paired-end sequencing was carried out on the HiSeqX Ten platform. Reads from all 991 individuals were aligned to the hg38 reference genome using BWA-MEM and duplicate reads were marked using Picard. Local realignment of insertion/deletions (INDELs) and base quality score recalibration were then performed using the Genome Analysis Toolkit version 3.4 (GATK) followed by SNV/INDEL calling with Haplotype Caller and Genotype GVCFs. VerifyBAMID (*Jun et al., 2012*) was used to estimate sample contamination. Samples with low coverage (<90% of genome covered at 10X) or a high contamination estimate (FREEMIX > 0.03) were excluded. A single VCF file with SNV and INDEL calls from all 991 individuals was created for subsequent statistical analysis.

## Rare variant burden analysis of *LRP2* and *APOB*

WGS data from cases and controls was compared for rare variant burden of the candidate genes that have been functionally validated in both systems (*LRP2, APOB*) (*Supplementary file 6*).

Genotypes with genotype quality (GQ) <20 were excluded, and the resulting data was used to calculate variant call rates and Hardy-Weinberg Equilibrium (HWE) p-values. Variants with call rate < 0.95 or HWE p-value<1e-8 were excluded. In addition, variants were required to pass VQSR (*McKenna et al., 2010*; *DePristo et al., 2011*; *Van der Auwera et al., 2013*). Variants were only included in the analysis if they had a strong predicted functional impact based on annotation information from Clinical Annotation of Variants (CAVA) (*Münz et al., 2015*). Specifically, we included frameshift, nonsynonymous, stop-gain, and stop-loss variants, as well as variants that alter an essential splice site. We further restricted the nonsynonymous variants to include only those with Combined Annotation Dependent Depletion (CADD) scores > 24 (*Kircher et al., 2014*). Rare variants (MAF <0.01 across all races) were identified based on allele frequencies in ExAC, gnomAD, and ESP (WES data from 6503 individuals, URL: http://evs.gs.washington.edu/EVS) (*Lek et al., 2016*). The gene-level, case-control association analysis was conducted using SKAT-O (*Lee et al., 2012*). Variants were weighted using the beta(1, 25) density function of the observed MAF (the default option in SKAT) and were mapped to genes using HG38 gene coordinates from Ensembl (*Frankish et al., 2017*). Correcting for multiple testing, the threshold for statistical significance was set at p<0.025 (0.05/2 genes).

After enrichment of rare, predicted-damaging missense variants in *LRP2* was established, we accounted for the potential influence of race and also relaxed functional constraints. Subsequent analyses were confined to 117 individuals with HLHS possessing > 80% of ancestral Caucasian alleles. All variants residing in the gene body of *LRP2* were included, in addition to 1000 base pairs upstream of the transcription start site. Variants were isolated and annotated in CAVA utilizing the canonical transcript of *LRP2* (ENST00000263816). In addition to analyzing the total number of variants spanning the gene body, SKAT-O analysis was performed separately for each type of variant in the following categories: missense, intronic, splice site region, splice region (in-frame, missense, synonymous), synonymous, 3' untranslated region, 5' untranslated region and 1000 base pairs upstream of the transcription start site. Independent of CAVA annotation, SKAT-O analysis was also performed on regulatory regions as determined by ChIP-Seq data from two different sources. The first analysis included variants within regions of *LRP2* impacted by histone modification and CTCF binding from publicly available ENCODE datasets (*Narayanan et al., 2017*). Twenty-one human cardiovascular tissues were assessed prior to confining the analysis to fetal human heart tissue (n = 3) (*Supplementary file 9*). The second analysis was confined to ENCODE ChIP-Seq data for 161 transcription factors in 91 cell types (wgEncodeRegTfbsClusteredV3 table in UCSC) (http://genome.ucsc.edu/).

## iPSC production and spontaneous differentiation of proband/parent cells

Fibroblasts were extracted from tissue by migration onto culture plates in fibroblast medium (DMEM, 10% Fetal bovine serum (FBS), penicillin/streptomycin (P/S), all from Thermo Fisher, Waltham, MA). For the reprogramming process, $5 \times 10^4$ fibroblasts were plated and incubated overnight to allow attachment as previously described (*Folmes et al., 2013*). On the infection day, medium was supplemented with lentivirus encoding reprogramming genes *SOX2, OCT4, KLF4*, and *c-MYC* and incubated for 12 hr. Cells were grown in fibroblast medium for 3 days prior to being passaged onto a matrigel coated plate. Once cells were attached, fibroblast medium was substituted by pluripotency-sustaining medium supplemented with 10 µM of ROCK inhibitor (TOCRIS, Bio-Techne, Minneapolis, MN) and refreshed daily until colonies appeared (3–6 weeks). Individual colonies were manually picked and expanded on matrigel coated plates in mTeSR1 medium (STEMCELL Technologies, Vancouver, CA). Approximately every 5–6 days cells were mechanically passaged onto fresh matrigel coated plates.

For spontaneous differentiation cells were treated with collagenase IV (Invitrogen, ThermoFisher) for 20 min, gently dislodged from the plate and transferred into suspension culture in ultra-low attachment 6-well plates in differentiation medium (DMEM/F12, 20% FBS, 1% glutamax, 1% nonessential amino acids, and 0.1% 2-mercaptoethanol). On day 5, floating aggregates were transferred to gelatin-coated tissue culture plates where medium was refreshed every 2 to 3 days. Cells were harvested for RNA extraction on day 25.

## iPSC characterization of proband/parent cells

For pluripotency characterization, cells were fixed in 4% paraformaldehyde for 10 min, permeabilized with 0.1% triton-X, blocked using Superblock and stained for membrane antigens TRA-1–60 (monoclonal mouse IgM 1:100), SSEA3 (rat 1:100) and transcription factor Nanog (rabbit 1:100) (all from Stemgent, Cambridge, MA). Characterization of sarcomeric proteins included staining for MLC2a (monoclonal mouse IgG 1:200, Synaptic Systems, Göttingen, Germany) and MLC2v (rabbit 1:200, Proteintech, Rosemont, IL). Secondary staining consisted of Alexa fluor 568 anti-mouse IgM or IgG, Alexa fluor 488 anti-rat and Alexa fluor 633 or 488 anti-rabbit, all used at 1:250 dilution (Molecular Probes, Thermo Fisher) (*Folmes et al., 2013*). Nuclei were stained with 4',6-diamidino-2-phenylindole (DAPI). Confocal images were acquired with a Zeiss LSM 510.

Pluripotency properties were determined in vivo using a teratoma assay. All studies including animals were approved by the Institutional Animal Care and Use Committee at Mayo Clinic. Half a million cells in 50 μl of a 1:1 solution of differentiation medium and matrigel were injected subcutaneously in each flank of an immunodeficient mouse. Tumor growth was monitored for up to 10 weeks with growing masses harvested as they reached a 1 cm$^3$ volume. Tissue was flash frozen, cryosectioned and stained using hematoxylin/eosin (*Folmes et al., 2013*).

Electron microscopy images were acquired with a JEOL 1200 EXII transmission electron microscope. Cells were processed through fixation with 1% glutaraldehyde and 4% formaldehyde in 0.1 M phosphate buffered saline (pH 7.2), staining with lead citrate and ultramicrotome sectioning prior to imaging (*Martinez-Fernandez et al., 2010*).

## Transcriptome profiling with RNA sequencing (RNA-seq) and bioinformatics analysis

RNA was extracted from iPSCs and differentiated cells at days 0 and 25 using a combination of Trizol and QIAGEN RNeasy mini kit columns. Sequencing library was prepared using TruSeq RNA Library Preparation Kit v2. All samples were sequenced on Illumina Hiseq 2000 at Mayo Clinic Medical Genome Facility. The following RNA-seq data analysis was performed on Dell Precision T7500 workstation which has 96 GB RAM and 20 Intel Xeon X5680 processors (3.33 GHz) and runs 64-bit Red Hat Enterprise Linux 6.3 (Kernel Linux 2.6.32–279.14.1.el6.x86_64). The alignment of short reads (50 bp) from FASTQ files was performed using Bowtie2 and Tophat2 software. All mapped reads were assembled to transcripts using Cufflinks and merged together using Cuffmerge. Differential analyses were performed between proband and parents at each time point (day 0 and day 25: d0, d25). The results from Cuffdiff were imported into a SQL database using R package CummeRbund for extracting significantly differential genes and other data manipulation. Differential genes were selected based on the default setting in Cuffdiff with adjusted p-values at 0.05 after FDR control for correcting multiple hypothesis tests and a minimum fold change of ± fold or greater relative to control lines. Bioinformatics analysis of gene expression changes was performed using available online tools to describe differential patterns between proband, mother and father. Gene functional annotation and classification was generated using the Database for Annotation, Visualization and Integrated Discovery bioinformatics module (http://david.abcc.ncifcrf.gov). Additionally, mapping was performed using the Kyoto Encyclopedia of Genes and Genomes array tool (http://www.kegg.jp/kegg/download/kegtools.html). Heat maps were generated from sorted Database for Annotation, Visualization and Integrated Discovery and Kyoto Encyclopedia of Genes and Genomes gene subsets using TIGR's open source MeV software (http://tm4.org/mev). In each sample, for each mapped gene, sample data points were normalized to the mean expression across proband, father and mother and subsequently log2 transformed. Significant function groups were ranked based on statistical significance (p) from hypergeometric distribution.

## Guided cardiac differentiation

Guided differentiation was achieved using a modified version of a previously published protocol (*Lian et al., 2013*). In brief, iPSCs cells were cultured as monolayer for two passages prior to induction. Next, they were treated with 8–12 μM Wnt activator CHIR99021 (Stemcell technologies) for 20 hr followed by 24 hr wash out period in DMEM:F12 with B27 supplement (Gibco, ThermoFisher). Medium was then refreshed and supplemented with 5 μM Wnt inhibitor IWP2 for 48 hr. Cells were maintained in DMEM:F12 plus B27 for an extra 48 hr and in DMEM:F12 plus B27 (minus

insulin) thereafter. Cultures were sampled for RNA extraction before induction as well as at days 1, 3, 5, 7, 14, and 37. Beating could be observed after 7–10 days of differentiation.

hiPSCs, hiPSC-CMs, siRNA transfection, EdU assay, Immunostaining and qRT-PCR hiPSCs derived from HLHS families were plated in 384 wells coated with matrigel at 5000 cell/well density using mTeSR-1 (Stem Cell). After 3 days, EdU was added to the media and was incubated for 1 hr. Cells were fixed in 4% PFA and stained for EdU and DAPI (Invitrogen). EdU was detected using Click-it Plus EdU Imaging Kit (Life Technologies) following manufacturing directions. iPSC-CMs from 5H, 75H and 151H families and healthy hiPSC-CMs (*Burridge et al., 2014*) at day 25 of differentiation were plated in 384 wells coated with matrigel at 5000 cells/well density in Maintenance Media (MM) (RPMI, 2% KOSR, 1% B27, 1% P/S). siRNA transfection was performed using Opti-Mem (Gibco) and Lipofectamine RNAiMAX (Gibco). siRNAs were purchased at Dharmacon and used at a final concentration of 25 nM. siRNA transfection efficiency was tested with qRT-PCR (*Figure 2—figure supplement 4A*). siRNA scramble was used as control (siCTR). To test WNT-pathway interaction, cells were treated with 1 uM BIO (GSK-3 inhibitor) (Sigma B1686) for three days. Two days after transfection, 50% of media was removed and replaced with 20 µM EdU in MM media. 24 hr after, cells were fixed in 4% paraformaldehyde (PFA) and blocked in blocking buffer (10% Horse Serum, 10% Gelatin, 0.5% Triton X-100). Cells were stained with mouse monoclonal anti-α-Actinin (ACTN1) (Sigma, A7811 1:800), secondary antibody Alexa fluor 568 anti-mouse (Invitrogen, 1:1000) and DAPI (1:1000) in blocking buffer and imaged using ImageXPress microscope, (Molecular Devices) and analyzed with MetaXpress Analysis software (Molecular Devices). To obtain a cardiomyocyte proliferation index, the total number of cells positive for EdU, α-Actinin and DAPI was divided by the total number of DAPI cells and expressed as percentage. For qRT-PCR experiments, total RNA was extracted using TRIzol and chloroform. 1 ug of RNA was converted in cDNA using QuantiTect Reverse Transcription kit (QIAGEN). qRT-PCR was performed using Syber green (Biorad). Human primers sequences for qRT-PCR were obtained from Harvard Primer Bank. TP53 (Primer Bank ID: 371502118 c1), CDKN1A (Primer Bank ID: 310832423 c1), LRP2 (Primer Bank ID: 126012572 c1), FZD10 (Primer Bank ID: 314122154 c3), PTCH1 (Primer Bank ID: 134254431 c3), CCNE1 (Primer Bank ID: 339275820 c3) PCNA (Primer Bank ID: 33239449 c1), CCNB1 (Primer Bank ID: 356582356 c1), CCNB2 (Primer Bank ID: 332205979 c1), CDK1 (Primer Bank ID: 281427275 c1), CRADD (Primer Bank ID: 51988883 c1), CASP6 (Primer Bank ID: 73622127 c1), CDKN2C (Primer Bank ID: 17981697 c1), CDKN1C (Primer Bank ID: 169790898 c1), APOB (Primer Bank ID: 105990531 c1), ELF4 (Primer Bank ID: 187608766 c1), HN1(JPT1) (Primer Bank ID: 7705877a1), HS6ST2 (Primer Bank ID: 116295253 c2), HSPG2 (Primer Bank ID: 140972288 c1), PRTG (Primer Bank ID: 224500891 c2), SDHD (Primer Bank ID: 222352156 c3), SIK1 (Primer Bank ID: 116256470 c1), SLC9A1 (Primer Bank ID: 381214343 c3). GAPDH (Primer Bank ID: 378404907 c1) was used as housekeeping gene and used to normalize the data. At least three independent biological replicates were performed for each experiment.

Quantitative Real Time PCR (qRT-PCR) in iPSC qRT-PCR for pluripotency and disease-associated markers was performed in iPSC samples. RNA was extracted using a combination of Trizol and QIAGEN RNeasy mini kit columns. cDNA for pluripotency assessment was synthesized using reverse transcriptase supermix reagents (Invitrogen, Thermo Fisher). In the case of expression levels during a time course of differentiation, a Biorad (Hercules, CA) iScript synthesis kit was used. qRT-PCR was performed using pre-designed primers (see key resources table). All values were normalized to *GAPDH*.

### *Drosophila* and zebrafish heart function studies

*Drosophila* orthologs were determined using the DIOPT database (*Hu et al., 2011*), and RNAi lines were obtained from the Vienna *Drosophila* Resource Center (VDRC) stock center and crossed to the heart-specific *Hand*$^{4.2}$-Gal4 driver alone or in combination with one copy of the *tinman* loss-of-function allele *tin*$^{346}$ (*Azpiazu and Frasch, 1993*). Fly hearts were filmed and analyzed according to standard protocol (*Fink et al., 2009*). In zebrafish, gene expression was manipulated using standard microinjection of morpholino (MO) antisense oligonucleotides (*Westerfield, 1993*). In addition, we performed targeted mutagenesis using CRISPR/Cas9 genome editing (*Talbot and Amacher, 2014*; *Gagnon et al., 2014*; *Irion et al., 2014*), to create insertion/deletion (INDEL) mutations in the *lrp2a* gene (F0). Zebrafish were raised to 72 hr post fertilization (hpf), immobilized in low melt agarose and the hearts were filmed and analyzed, as for *Drosophila* (*Fink et al., 2009*).

## Zebrafish husbandry

All zebrafish experiments were performed in accordance to protocols approved by IACUC. Zebrafish were maintained under standard laboratory conditions at 28.5℃. In addition to Oregon AB wild-type, the following transgenic lines were used: *Tg(myl7:EGFP)*[twu277] (*Huang et al., 2003*) and *Tg (myl7:H2A-mCherry)*[sd12] (*Schumacher et al., 2013*).

## Zebrafish semi-automated optical heartbeat analysis (SOHA)

Larval zebrafish (72 hpf) were immobilized in a small amount of low melt agarose (1.5%) and sub-merged in conditioned water. Beating hearts were imaged with direct immersion optics and a digital high-speed camera (up to 200 frame/s, Hamamatsu Orca Flash) to record 30 s movies; images were captured using HC Image (Hamamatsu Corp.). Cardiac function was analyzed from these high-speed movies using semi-automatic optical heartbeat analysis software (*Fink et al., 2009*; *Ocorr et al., 2009*), which for zebrafish quantifies heart period (R-R interval), cardiac rhythmicity, as well as chamber size and fractional area change. All hearts were imaged at room temperature (20–21℃). Statistical analyses were performed using Prism software (Graphpad). Significance was determined using two-tailed, unpaired Student t-test or one-way ANOVA and Dunnett's multiple comparisons post hoc test as appropriate.

## Zebrafish cardiomyocyte cell counts and cardiac immunofluorescent imaging

To count cardiomyocytes, we used the expression of H2AmCherry in the nuclei (*Tg(myl7:H2A-mCherry)*) (*Schumacher et al., 2013*) to qualify as an individual cell, performed the 'Spot' function in Imaris to distinguish individual cells in reconstructions of confocal z-stacks (*Zeng and Yelon, 2014*; *Pradhan et al., 2017*). To compare data sets, we used Prism software (GraphPad) to perform Student's t-test with two-tail distribution. Graphs display mean and standard deviation for each data set.

Whole-mount immunofluorescence was performed as previously described (*Zeng and Yelon, 2014*; *Pradhan et al., 2017*; *Alexander et al., 1998*) (see key resources table). Confocal imaging was performed on an LSM 710 confocal microscope (Zeiss, Germany) with a 40x water objective. Exported z-stacks were processed with Imaris software (Bitplane), Zeiss Zen, and Adobe Creative Suite software (Photoshop and Illustrator 2020). All confocal images shown are projection views of partial reconstructions from multiple z-stack slices, except where noted that images are views of a single slice.

## Zebrafish CRISPR/Cas9 experiments

Detailed steps for *lrp2a* were previously described (*Hoshijima et al., 2019*) and we followed IDT manufacture instruction for Complexes preparation.

### crRNA:tracrRNA Duplex Preparation

Target-specific Alt-R crRNA (Dr.Cas9.LRP2A.1.AC,/AltR1/rCrC rCrUrC rGrCrU rUrArU rArUrU rCrUrC rCrArA rGrUrU rUrUrA rGrArG rCrUrA rUrGrC rU/AltR2/) and common Alt-R tracrRNA were synthesized by IDT and each RNA was dissolved in duplex buffer (IDT) as 100 µM stock solution. Stock solutions were stored at −20℃. To prepare the crRNA:tracrRNA duplex, equal volumes of 100 µM Alt-R crRNA and 100 µM Alt-R tracrRNA stock solutions were mixed together and annealed by heating followed by gradual cooling to room temperature by manufacture instruction: 95℃, 5 min on PCR machine; cool to 25℃; cool to 4℃ rapidly on ice. The 50 µM crRNA:tracrRNA duplex stock solution was stored at −20℃.

### Preparation of crRNA:tracrRNA:Cas9 RNP Complexes

Cas9 protein (Alt-R S.p. Cas9 nuclease, v.3, IDT) was adjusted to 25 µM stock solution in 20 mM HEPES-NaOH (pH 7.5), 350 mM KCl, 20% glycerol, dispensed as 8 ul aliquots, and stored at −80℃. 25 µM crRNA:tracrRNA duplex was produced by mixing equal volumes of 50 µM crRNA:tracrRNA duplex stock and duplex buffer (IDT). We used 5 µM RNP complex. To generate 5 µM crRNA:tracrRNA:Cas9 RNP complexes: 1 µl 25 µM crRNA:tracrRNA duplex was mixed with 1 µl 25 µM Cas9 stock, 2 µl H$_2$O, and 1 µl 0.25% phenol red solution. Prior to microinjection, the RNP complex

solution was incubated at 37℃, 5 min and then placed at room temperature. Approximately, one nanoliter of 5 µM RNP complex was injected into the cytoplasm of one-cell stage embryos to generate F0 larva.

## Statistical analysis

The qPCR time course gene expression data were analyzed using Generalized Linear Model (GLM) to assess the statistical significance. EdU-incorporation experiments and pTP53 staining were analyzed with GraphPad Prism 8. For both, $p < 0.05$ was considered significant. All statistical analysis for iPSC-derived cardiomyocytes were performed using GraphPad Prism version 8.0 (GraphPad Software, San Diego CA, USA). Statistical significance was analyzed by unpaired Student's *t*-test, and one-way ANOVA and shown as mean ± SEM. P-values were considered significant when $p < 0.05$.

## Study limitations

HLHS candidate gene selection was based on in silico predictive algorithms to filter for functional coding and regulatory variants. Our WGS filtering strategy, designed to identify major-effect de novo, recessive and loss-of-function variants, did not include consideration of inherited, incompletely penetrant, autosomal dominant variants in other genes. The potential race-specific differences in *LRP2* variants require further study. Differential gene expression, which was functionally validated as a powerful filter for candidate variant prioritization, excluded functional variants that do not alter gene expression. The validating KD modeling systems are justified insofar as all 10 prioritized candidate genes harbored recessive alleles inherited from the proband's unaffected parents, implicating a loss-of-function mechanism is likely in most cases. Not all human genes are conserved in *Drosophila*, but ~ 80% of disease-causing human genes have fly orthologs. While structural differences exist between hiPSC-CM, *Drosophila* and zebrafish hearts and human newborn cardiomyocytes, our combinatorial approach allows to uncover testable gene networks and interactions that is not feasible in mammalian model systems.

# Acknowledgements

The authors acknowledge the generous support from the Todd and Karen Wanek Family Program for Hypoplastic Left Heart Syndrome and the Medical Genome Facility Sequencing Core and Biobank within the Mayo Clinic Center for Individualized Medicine, Rochester, MN. We gratefully acknowledge the patients and families who participated in this study. We thank Sean Spearing, Prashila Amatya, Marco Tamayo and Bosco Trinh for excellent technical assistance.

# Additional information

### Funding

| Funder | Grant reference number | Author |
|---|---|---|
| Todd and Karen Wanek Family Program for Hypoplastic Left Heart Syndrome at Mayo Clinic Foundation | SAN-233970 | Alexandre R Colas<br>Rolf Bodmer |
| National Institutes of Health | HL054732 | Rolf Bodmer |

The funders had no role in study design, data collection and interpretation, or the decision to submit the work for publication.

### Author contributions

Jeanne L Theis, Conceptualization, Formal analysis, Validation, Investigation, Visualization, Methodology, Writing - original draft, Writing - review and editing; Georg Vogler, Conceptualization, Data curation, Formal analysis, Supervision, Validation, Investigation, Visualization, Methodology, Writing - original draft, Writing - review and editing; Maria A Missinato, Conceptualization, Data curation, Formal analysis, Supervision, Validation, Investigation, Visualization, Methodology, Writing - original draft, Project administration, Writing - review and editing; Xing Li, Conceptualization, Resources,

Data curation, Software, Formal analysis, Investigation, Methodology, Writing - original draft, Writing - review and editing; Tanja Nielsen, Formal analysis, Investigation, Methodology; Xin-Xin I Zeng, Anaïs Kervadec, James N Kezos, Formal analysis, Investigation, Visualization, Methodology; Almudena Martinez-Fernandez, Conceptualization, Formal analysis, Investigation, Methodology, Writing - original draft; Stanley M Walls, Formal analysis, Investigation, Visualization; Katja Birker, Formal analysis, Investigation; Jared M Evans, Megan M O'Byrne, Zachary C Fogarty, Data curation, Software, Formal analysis, Methodology; André Terzic, Conceptualization, Resources, Supervision, Project administration; Paul Grossfeld, Conceptualization, Resources, Supervision, Investigation; Karen Ocorr, Conceptualization, Software, Formal analysis, Supervision, Funding acquisition, Investigation, Methodology, Writing - review and editing; Timothy J Nelson, Conceptualization, Resources, Supervision, Funding acquisition, Writing - original draft, Project administration, Writing - review and editing; Timothy M Olson, Conceptualization, Resources, Formal analysis, Supervision, Investigation, Methodology, Writing - original draft, Project administration, Writing - review and editing; Alexandre R Colas, Conceptualization, Resources, Supervision, Funding acquisition, Investigation, Methodology, Writing - original draft, Project administration, Writing - review and editing; Rolf Bodmer, Conceptualization, Resources, Formal analysis, Supervision, Funding acquisition, Investigation, Writing - original draft, Project administration, Writing - review and editing

## Author ORCIDs

Georg Vogler ⓘ https://orcid.org/0000-0002-8303-3531
Maria A Missinato ⓘ http://orcid.org/0000-0001-9055-758X
Xin-Xin I Zeng ⓘ http://orcid.org/0000-0002-2707-7759
Zachary C Fogarty ⓘ http://orcid.org/0000-0001-5588-3216
Rolf Bodmer ⓘ https://orcid.org/0000-0001-9087-1210

## Ethics

Human subjects: Written informed consent was obtained for the index family and an HLHS cohort, under a research protocol approved by the Mayo Clinic Institutional Review Board (11-000114 "Genetic Investigations in Hypoplastic Left Heart Syndrome").

Animal experimentation: SBP has retained the services of a veterinarian who is a diplomat of the American College of Laboratory Animal Medicine. Close contact with Animal Facility personnel is maintained through telephone calls and on-campus visits once a week. This person is a member of the Institute's Animal Care and Use Committee (IACUC)and attends monthly meetings. This study was performed in strict accordance with the recommendations in the Guide for the Care and Use of Laboratory Animals of the National Institutes of Health. All of the animals were handled according to the Institute's Animal Care and Use Program, which is accredited by AAALAC International, and a Multiple Project Assurance A3053-1 is on file in the OLAW, DHHS. The protocol was approved by SBP IACUC (Permit Number: 19-087). Animals are euthanized after filming hearts by an overdose of anesthetic (3-aminobenzoic acid ethyl ester (MS-222)) at 250-300 mg/L.

## Decision letter and Author response

Decision letter https://doi.org/10.7554/eLife.59554.sa1
Author response https://doi.org/10.7554/eLife.59554.sa2

# Additional files

## Supplementary files

• Supplementary file 1. Differentially expressed genes between proband and parental 25 day old iPSC-derived cardiomyocytes.

• Supplementary file 2. Gene function enrichment analysis for differentially expressed transcripts at day 25.

• Supplementary file 3. Recessive and dominant genotypes identified in index case.

• Supplementary file 4. Differentially expressed genes between proband and parental 0 day old iPSCs.

- Supplementary file 5. Interactome of prioritized HLHS candidate genes.
- Supplementary file 6. SKAT-O Analysis of 2 prioritized candidate genes in cases and controls.
- Supplementary file 7. Rare, predicted-damaging LRP2 missense variants in cases and controls (CADD>24).
- Supplementary file 8. Case-Control Association Analysis of Rare Variants in LRP2.
- Supplementary file 9. ENCODE Datasets.
- Transparent reporting form

### Data availability

Sequencing data are deposited in the NCBI Sequence Read Archive (SRA) database with accession numbers: SRS1417684 (proband iPSCs), SRS1417685 (paternal iPSCs), SRS1417686 (maternal iPSCs), SRS1417695 (proband d25 differentiated cells), SRS1417696 (paternal d25 differentiated cells), SRS1417714 (maternal d25 differentiated cells).

The following previously published dataset was used:

| Author(s) | Year | Dataset title | Dataset URL | Database and Identifier |
|---|---|---|---|---|
| Mayo Clinic | 2017 | IPS Cells in Hypoplastic Left Heart Syndrome | https://www.ncbi.nlm.nih.gov/sra/SRS1417684 | NCBI Sequence Read Archive, SRX1736972 |

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
