## [Decision Letter]

**Acceptance summary:**

This paper describes a new method platform to facilitate understanding of the molecular basis of complex heart diseases. Employing a combined analysis of human, zebrafish, and Drosophila model systems, this study identifies and characterizes an exemplary genetic variant relevant to the development of Hypoplastic Left Heart Syndrome (HLHS). Use of this technological tool could facilitate future basic as well as translational research on the development of heart diseases.

**Decision letter after peer review:**

Thank you for submitting your article "Patient-specific functional genomics and disease modeling suggest a role for *LRP2* in hypoplastic left heart syndrome" for consideration by *eLife*. Your article has been reviewed by three peer reviewers, one of whom is a member of our Board of Reviewing Editors, and the evaluation has been overseen by Didier Stainier as the Senior Editor. The reviewers have discussed the reviews with one another, and the Reviewing Editor has drafted this decision to help you prepare a revised submission.

Summary:

Hypoplastic Left Heart Syndrom (HLHS) is a severe congenital heart disease which remains genetically incompletely understood. The authors use a cross-functional genetic approach and a combination of *Drosophila*, zebrafish, and human iPSC model systems to identify novel genetic elements that underlie HLHS. The authors selected a variant in LDL receptor related protein 2 (*LRP2*) identified from a sporadic HLHS proband-parent trio to examine contribution of this gene defect to HLHS disease phenotypes. WGS data were filtered for rare de-novo, recessive and loss-of-function variants in candidate genes prior to functional evaluation. Three different model systems were utilized to evaluate dysregulations in the presence of the selected *LRP2* variant in context of HLHS. This integrated analysis is powerful and could indeed help in identifying and assessing genetic determinants of human disorders.

The authors suggest a model in which Wnt and Shh-mediated signaling impairs cardiomyocyte proliferation in HLHS. New evidence provided to support this model is scarce; the mechanisms of FZD10/PTCH1-dependent effects on *LRP2* are not sufficiently substantiated. Also, there are concerns regarding assessment of the cardiomyocyte proliferation defect.

Overall, this manuscript addresses a topic of interest with state of the art methodology and represents a valuable tool and resource. As a resource platform, this tool could be of broader use to researchers in the field. We have the following suggestions to improve the manuscript.

Essential revisions:

1) To lock in mechanistic evidence for the proposed model would require substantial new experimental data. On the other hand, the integrated approach employed in this manuscript contributes merit as a resource platform. To be considered as a tool and resource, conclusions should be adapted to reflect that the novel contribution of this paper is a technological platform which may identify novel variant mutations in genetic disorders such as HLHS. Claims on the presented network model as a mechanistic basis, as well as on causality of the *LRP2* variant should be removed or reduced. The authors should especially re-consider the context of the existing *LRP2* mutant mouse model (Baardman et al., 2016) which displays a left-ventricular non-compaction phenotype. Moreover, the authors should discuss how their experimental pipeline could be extended to better assess causality of the *LRP2* variant.

2) Experiments addressing the function of *LRP2* as an HLHS candidate gene using adult Drosophila hearts are not convincing, as postmitotic cardiac cells will limit an assessment of proliferation defects. The effect of *LRP2* on cardiac cell proliferation should be better documented. For example, testing candidate gene function on cardioblast formation during heart development in Drosophila embryos or, alternatively, zebrafish embryos could be more informative.

3) The title should indicate this work to cover a cross-species genomics method platform rather than disease modeling.

Revisions expected in follow-up work:

1) A causal link that the *LRP2* variant mutation critically contributes to HLHS disease phenotypes should be provided, using an *LRP2* variant Drosophila or zebrafish line, or mutation-correction in patient-specific iPSCs.

2) Experiments should be performed to demonstrate how *LRP2* regulates cell cycle and cardiomyocyte proliferation via Wnt/Shh-dependent signaling. Can a direct *LRP2*-dependent regulation of FZD10/PTCH1 be confirmed e.g. in interaction studies? To support modulation by Wnt agonists/Shh antagonists in presence of the *LRP2* variant, additional regulators of these pathways should be as assessed. The zebrafish model (CRISPR *lrp2a* mutants) or an *LRP2* variant Drosophila line could be used for this together with siRNAs or CRISPR for knock-down/ knockout, targeting *LRP2*.

---

## [Author Response]

We would like to draw your attention to changes in our revision policy that we have made in response to COVID-19 (https://elifesciences.org/articles/57162). Specifically, when editors judge that a submitted work as a whole belongs in eLife but that some conclusions require a modest amount of additional new data, as they do with your paper, we are asking that the manuscript be revised to either limit claims to those supported by data in hand, or to explicitly state that the relevant conclusions require additional supporting data.

We thank the reviewers and editors for their positive evaluation of our work. As requested, we have revised the manuscript to limit claims the reviewers deemed insufficiently supported by our data, and where appropriate, we have explicitly stated that additional lines of evidence are needed. Below, we summarized the planned additional experiments over the next months (COVID-19 situation permitting) that we hope will provide additional data in support of our conclusions.

Our expectation is that the authors will eventually carry out the additional experiments and report on how they affect the relevant conclusions either in a preprint on bioRxiv or medRxiv, or if appropriate, as a Research Advance in eLife, either of which would be linked to the original paper.

Summary of ‘additional experiments’ that are planned (resulting data are intended to be reported in a self-standing document submitted to Research Advance in *eLife*, and/or uploaded to the BioRxiv server, and linked to the original paper):

– Genetic interaction studies between *LRP2/mgl* and component of SHH/WNT pathways in *Drosophila,* hiPSC-CM, and best subset in zebrafish

– Experiment addressing *LRP2* effect on proliferation: We will use zebrafish to examine loss function of *lrp2a* (and key potential interactors) on proliferation in EdU incorporation assays.

Summary:Hypoplastic Left Heart Syndrom (HLHS) is a severe congenital heart disease which remains genetically incompletely understood. The authors use a cross-functional genetic approach and a combination of Drosophila, zebrafish, and human iPSC model systems to identify novel genetic elements that underlie HLHS. The authors selected a variant in LDL receptor related protein 2 (LRP2) identified from a sporadic HLHS proband-parent trio to examine contribution of this gene defect to HLHS disease phenotypes. Whole-genome sequencing data were filtered for rare de-novo, recessive and loss-of-function variants in candidate genes prior to functional evaluation. Three different model systems were utilized to evaluate dysregulations in the presence of the selected LRP2 variant in context of HLHS. This integrated analysis is powerful and could indeed help in identifying and assessing genetic determinants of human disorders.

We thank the reviewers for their appreciation of our integrated analysis platform for assessing the cardiac role of HLHS disease gene candidates. We have further stressed the novelty of our synergistic approaches in gene discovery and toned-down conclusions relating to HLHS disease mechanisms (see text changes in track throughout the revised manuscript).

The authors suggest a model in which Wnt and Shh-mediated signaling impairs cardiomyocyte proliferation in HLHS. New evidence provided to support this model is scarce; the mechanisms of FZD10/PTCH1-dependent effects on LRP2 are not sufficiently substantiated. Also, there are concerns regarding assessment of the cardiomyocyte proliferation defect.

We appreciate this criticism. We will perform genetic interaction studies in our model systems to further evaluate a *LRP2*/*mgl*-Shh/Wnt connection, and further assess CM proliferation in zebrafish.

– Genetic interactions between *LRP2* and WNT/SHH components in iPSC-CMs: In Figure 4D-G of the manuscript, we reported KD of *LRP2* in combination with PTCH1 using siRNA, and *LRP2* KD in combination with the WNT activator drug ‘BIO’. To further test LRP2-WNT/SHH interactions, we will use siRNAs for additional Wnt/Shh pathway genes in hiPSC-CMs treated with siLRP2 alone and in combination of the other siRNAs (EdU incorporation assay).

– Genetic interactions between *mgl*/*LRP2* and *wg/hh* pathway components in Drosophila embryonic cardioblasts and adult heart function: To examine the effects of genetic interactions on embryonic cardioblasts, we will use early mesodermal and cardioblast drivers (e.g. *Mef2*-Gal4, *TinD*-Gal4, resp.) to KD *wg/hh* components in a *mgl* mutant and mid-cardioblast marked background (*mgl/Y;*Gal4*>hh/wg-RNAi,mid-GFP* – see Schroeder et al., 2019) – also an aim in a submitted grant proposal). Various antibodies will be used to determine abnormalities in subsequently fixed embryos. To examine interaction in adult hearts, we will KD *mgl* and interactor in the heart throughout life (e.g. *Hand^4.2^*-Gal4>*hh/wg-RNAi,tdtK-RFP*) and determine functional and structural defects. We have evaluated 800+ genes in a year using this platform.

– Genetic interactions between *LRP2* and Wnt/Shh components in zebrafish: Best interactors in hiPSC-CM proliferation assay and embryonic and adult fly heart assays will be tested in zebrafish first using co-injection of subthreshold doses of MO against *lrp2a* combined with MO of potential interactors. Interactions will then be also confirmed with CRISPR generated *lrp2a* mutants.

– The impact of LRP2 on proliferation will be further investigated zebrafish *lrp2a* morphants and CRISPR mutants, by proliferation analysis using EdU incorporation and in cell death assays.

Overall, this manuscript addresses a topic of interest with state of the art methodology and represents a valuable tool and resource. As a resource platform, this tool could be of broader use to researchers in the field. We have the following suggestions to improve the manuscript.Essential revisions:1) To lock in mechanistic evidence for the proposed model would require substantial new experimental data. On the other hand, the integrated approach employed in this manuscript contributes merit as a resource platform. To be considered as a tool and resource, conclusions should be adapted to reflect that the novel contribution of this paper is a technological platform which may identify novel variant mutations in genetic disorders such as HLHS. Claims on the presented network model as a mechanistic basis, as well as on causality of the LRP2 variant should be removed or reduced.

We agree, this paper described a novel integrated platform for evaluation heart disease candidates (e.g. from genomics and transcriptomics analysis of heart disease patients) and potential disease-relevant genetic interactions. We have toned down the conclusions that claim causality of *LRP2* in HLHS, such as identified *LRP2* variants contributing to the disease – see extensive track changes throughout the manuscript.

The authors should especially re-consider the context of the existing LRP2 mutant mouse model (Baardman et al., 2016) which displays a left-ventricular non-compaction phenotype.

The reviewer implies that the *LRP2* mutant mouse model yields solely a LVNC phenotype. Yet, on further inspection of Baardman et al. (Baardman et al., 2016), mice develop LVNC and CHDs. In the second paragraph of the subsection “Integrated multidisciplinary disease gene discovery platform”, we now discuss the possibility that *LRP2* could contribute to several CHDs, involving not just proliferation but also proper differentiation, such as LVNC). Of note, it is well known that core cardiogenic genes, like *NKX2-5*, are likely involved in several CHDs.

Moreover, the authors should discuss how their experimental pipeline could be extended to better assess causality of the LRP2 variant.

We also discuss in more detail (e.g. in the last paragraph of the subsection “Integrated multidisciplinary disease gene discovery platform”) how the pipeline could be used to further explore causality, such as stated below under Follow-up Work (2.).

2) Experiments addressing the function of LRP2 as an HLHS candidate gene using adult Drosophila hearts are not convincing, as postmitotic cardiac cells will limit an assessment of proliferation defects. The effect of LRP2 on cardiac cell proliferation should be better documented. For example, testing candidate gene function on cardioblast formation during heart development in Drosophila embryos or, alternatively, zebrafish embryos could be more informative.

*Drosophila* hearts not easily amenable to assay the role of genes in cardiomyocyte proliferation, since proliferation of cardioblasts is complete very early, before heart tube assembly. Rather, the embryonic heart is best used for determining cardiac cell specification, differentiation, and eventually establishment of robust cardiac function, the latter being a major concern in adult CHD. Complementary, as stated above for the genetic interaction experiments, we will also use cell-type specific markers to assess cardiac specification of the embryonic fly heart. Further, we will use zebrafish to better document *LRP2*’s role on cardiac cell proliferation (EdU incorporation).

3) The title should indicate this work to cover a cross-species genomics method platform rather than disease modeling.

We agree that it was not our intent to perform disease-proximal modeling, but rather perform a genetic evaluation of candidate genes using our integrated platform. We propose to change the title to "Patient-specific genomics and cross-species functional analysis implicate a role for *LRP2* in hypoplastic left heart syndrome".

Revisions expected in follow-up work:1) A causal link that the LRP2 variant mutation critically contributes to HLHS disease phenotypes should be provided, using an LRP2 variant Drosophila or zebrafish line, or mutation-correction in patient-specific iPSCs.2) Experiments should be performed to demonstrate how LRP2 regulates cell cycle and cardiomyocyte proliferation via Wnt/Shh-dependent signaling. Can a direct LRP2-dependent regulation of FZD10/PTCH1 be confirmed e.g. in interaction studies? To support modulation by Wnt agonists/Shh antagonists in presence of the LRP2 variant, additional regulators of these pathways should be as assessed. The zebrafish model (CRISPR lrp2a mutants) or an LRP2 variant Drosophila line could be used for this together with siRNAs or CRISPR for knock-down/ knockout, targeting LRP2.

We thank the reviewers for these suggestions. In fact, this is exactly what we have in mind in long-term follow-up studies. We have submitted an R01 grant proposal to NIH on 1) generating *LRP2* variants in our models and proliferation rescue by CRISPR/Cas9 correction of patient-derived iPSCs; and on 2) LRP2 regulating cell proliferation in cooperation with Wnt/Shh by genetic interaction studies, as well as other candidate genes with patient-specific rare, predicted-damaging variants prioritized in other HLHS family trios. These experiments, however, are a longer-term goal, requiring significant resources and efforts.